# Cost-Free Fairness in Online Correlation Clustering

**Eric Balkanski**                                                           EB3224@COLUMBIA.EDU
*Columbia University*

**Jason Chatzitheodorou**                                                    IC2621@COLUMBIA.EDU
*Columbia University*

**Andreas Maggiori**                                                         AM6292@COLUMBIA.EDU
*Columbia University*

**Editors:** Gautam Kamath and Po-Ling Loh

## Abstract

In the correlation clustering problem, the input is a signed graph where the sign indicates whether pairs of nodes should be placed in the same cluster or not. The goal is to create a clustering that minimizes the number of disagreements with these signs. Correlation clustering is a key unsupervised learning problem with many practical applications. It has been widely studied in various settings, including versions with fairness constraints and cases where nodes arrive online. In this paper, we explore a problem that combines these two settings: nodes arrive online and reveal their membership in protected groups upon arrival. We are only allowed to output fair clusters, i.e., clusters where the representation of each protected group is upper bounded by a user-specified constant at the beginning of the arrival sequence. Our aim is to maintain approximately optimal fair clustering while minimizing a node's worst-case recourse, i.e., the number of times it changes clusters. We present an algorithm that achieves worst-case logarithmic recourse per node while maintaining a constant-factor fair approximate clustering. Additionally, our approach simplifies the algorithm and analysis used in prior work by Cohen-Addad et al. (2022) in the online setting with recourse.
**Keywords:** fairness, correlation clustering, online algorithms

## 1. Introduction

The fairness of algorithms used to make decisions that directly affect individuals is an important concern in a growing number of applications, such as recidivism predictions for bail determinations (Dressel and Farid, 2018; Wadsworth et al., 2018), hiring processes (Raghavan et al., 2020; Schumann et al., 2020), and credit-risk assessment for loan approvals (Kumar et al., 2022; Kozodoi et al., 2022). Motivated by these applications, there has been extensive literature on designing fair algorithms for a wide variety of problems, including classical problems in machine learning and algorithm design such as resource allocation (Li et al., 2020; Bateni et al., 2022), classification (Agarwal et al., 2018; Zafar et al., 2017), and clustering (Chierichetti et al., 2017; Bera et al., 2019). The vast majority of this work has studied fairness in offline settings where the entire input is known to the algorithm. However, loan, bail, and hiring decisions are often made in an online manner. Satisfying fairness properties in an online setting is significantly more challenging than in offline settings because, in an online problem, the fairness constraints are evolving over time and decisions must be made without knowing what these constraints will be in the future.

In this paper, we study fair and online decision-making in the context of the classical correlation clustering problem. In this problem, the input is a complete graph where each edge is either marked positive, indicating two nodes are similar, or negative, indicating dissimilarity. The goal is to cluster

the nodes such that the number of intra-cluster negative edges and inter-cluster positive edges are minimized. An important application of correlation clustering is automated labeling (Agrawal et al., 2009; Chakrabarti et al., 2008), which is used in scenarios where it is expensive to individually label each point in a large data set. Instead, automated labeling clusters the points based on their features and points in the same cluster are then assigned the same label. Although automated labeling is an appealing approach to labeling massive datasets very quickly, important fairness concerns emerge when these labels lead to decision-making that directly impacts individuals.

A long line of work has studied fair clustering, including fair correlation clustering (Ahmadian et al., 2020; Ahmadi et al., 2020; Ahmadian and Negahbani, 2023; Schwartz and Zats, 2022; Friggstad and Mousavi, 2021), where each node has one, or more, colors that encode protected features. In the fairness setting from (Ahmadian and Negahbani, 2023), which is the fairness setting we consider, there is a parameter $a_i \in [0, 1]$ associated with color $i$ and a clustering is fair if each cluster either is a singleton[1] or it contains, for each color $i$, at most an $a_i$ fraction of nodes of color $i$. In this setting, when $a_i > 0$ and $\rho > 0$ are constants, Ahmadian and Negahbani (2023) achieve a clustering that is a constant factor approximation to the optimal fair solution and $\rho$-fair, meaning that it violates the fairness constraints by at most a $1 + \rho$ multiplicative factor.

Since data points often arrive online, there is a separate line of work that has studied online correlation clustering. In the online setting, nodes $u$ arrive one by one and, upon arrival, the sign marking of each edge between $u$ and a previously arrived node $v$ is revealed, and the algorithm must irrevocably decide which cluster node $u$ should be assigned to. Mathieu et al. (2010) have shown a strong $\Omega(n)$ lower bound on the best competitive ratio achievable for this online correlation clustering problem. A standard relaxation in online algorithms to overcome strong impossibility results is to consider recourse, where a small number of past decisions can be changed, which was initiated by Lattanzi and Vassilvitskii (2017) in the context of clustering. Cohen-Addad et al. (2022) give an algorithm for online correlation clustering that maintains a constant factor approximation at every time step and has $O(\log n)$ recourse per node, i.e., each node is re-assigned a cluster at most logarithmically many times. Due to the importance of both the online and fair settings for correlation clustering, our main question is the following.

*Is there a constant competitive algorithm for online correlation clustering*
*that is fair and has low recourse?*

Ideally, we would like to obtain the best-known fairness guarantee achieved by an offline constant-factor approximation algorithm as well as the best-known logarithmic recourse achieved by an online constant competitive algorithm. We emphasize that it is a priori unclear if such an algorithm exists since maintaining the fairness guarantees could potentially cause a large recourse. Our main result is that we answer the above question affirmatively.

**Theorem** *For the online correlation clustering problem, if $a_i, \rho > 0$ are constants for all $i$, then there is an algorithm that maintains, with $O(\log n)$ recourse per node, a solution that is, at every time step, a constant factor approximation and $\rho$-fair.*

This result shows that the best-known fairness and online guarantees can both be simultaneously achieved. Since Cohen-Addad et al. (2022) give a tight lower bound showing that $\Omega(\log n)$ recourse

---

1. One motivation for considering a singleton cluster to be fair is in the context of automated labeling, where the label of a node in a singleton cluster is not impacted by other nodes.

is required for maintaining a solution that is a constant approximation, our result implies that there is no cost in the recourse and competitive ratio guarantees when adding fairness constraints for online correlation clustering. Additionally, our approach simplifies the algorithm and analysis used in prior work by Cohen-Addad et al. (2022) in the online setting with recourse.

**Technical overview.**   The main technical challenge in our work lies in computing clusters that remain stable over successive iterations, regardless of how new nodes connect or their colors. In traditional online settings with recourse, the challenge is limited to maintaining low recourse under changing node connections, which is already complex. However, simple examples reveal that algorithms optimized for low recourse can result in highly unfair clusters. A key insight of our work (formally presented in Lemma 6) is that clusters violating the fairness condition beyond a tolerance $\rho$ can be split into singletons, with the incurred cost being charged to the optimal fair clustering. This means that splitting highly unfair clusters does not lead to excessive costs.

Building on this, we design a procedure that monitors the fairness of evolving clusters. If a cluster becomes significantly unfair (i.e., violating some fairness constraint by more than $1 + \rho_1$), the procedure splits it. If all fairness constraints are satisfied with a slack of at most $1 + \rho_2$, the cluster remains intact. In cases where neither condition is met, the procedure repeats the decision made in the previous iteration. By setting $\rho_1 = 2\rho_2 = \rho$, we ensure that the cost of maintaining a cluster can be bounded by creating a multiplicative gap between $1 + \rho_1$ and $1 + \rho_2$. This multiplicative gap ensures that when a cluster is split due to unfairness and then merged again, it is because its size has increased by a multiplicative factor, which can only occur logarithmically many times.

As a byproduct of our approach, we significantly simplify both the algorithm and the proof of Cohen-Addad et al. (2022) for the online setting with recourse.

## 2. Problem Statement and Preliminaries

**Correlation clustering.**   In correlation clustering, the input is a complete signed undirected graph $G^s = (V, E, s)$ where each edge $e = \{u, v\}$ is assigned a sign $s(e) \in \{`+', `-'\}$. To ease notation, we convert this complete signed graph $G^s = (V, E, s)$ to a non-signed graph $G = (V, E)$ where for each pair of nodes $\{u, v\}$ there is an edge between them in $G$ if and only if $s(\{u, v\}) = `+'$ in $G^s$. Using this conversion and the standard *disagreements minimization* objective, the quality of a partition $\mathcal{C} = \{C_1, C_2, \dots\}$ of $V$, i.e., a clustering, is:

$$\text{cost}(\mathcal{C}) = |(u, v) \in E : u \nsim_{\mathcal{C}} v| + |(u, v) \notin E : u \sim_{\mathcal{C}} v|$$

where $u \sim_{\mathcal{C}} v$ if and only if $\exists C_i \in \mathcal{C}$ such that $u, v \in C_i$ and $u \nsim_{\mathcal{C}} v$ otherwise.

**Fairness.**   In fair correlation clustering, each node $u$ is assigned at least one of $l$ colors where the sets $V_1, \dots, V_l \subseteq V$ correspond to those colors. A clustering $\mathcal{C}$ is called fair with respect to the fairness parameters $a_1, \dots, a_l$ if and only if for every $C \in \mathcal{C}$:

$$\text{(fair)} \qquad |C \cap V_i| \leq a_i |C| \; \forall i \qquad \text{or} \qquad |C| = 1$$

Following (Ahmadian and Negahbani, 2023), we consider a singleton cluster $C$ (i.e., $|C| = 1$) to be fair. This aligns with the earlier discussion of automated labeling, where a node clustered as a singleton is unaffected by other nodes' labels. Consistent with (Ahmadian and Negahbani, 2023), we define a $\rho$-fair clustering as one where fairness constraints are violated by at most a factor of

$(1 + \rho)$. Formally, a clustering $\mathcal{C}$ is $\rho$-fair with respect to fairness parameters $a_1, \ldots, a_l$ if and only if, for every $C \in \mathcal{C}$:

$$(\rho\text{-fair}) \qquad |C \cap V_i| \leq (1 + \rho)a_i|C| \; \forall i \qquad \text{or} \qquad |C| = 1$$

**Online arrival.** In online correlation clustering, nodes arrive one at a time, revealing upon arrival its colors and all the edges to previously arrived nodes. An instance of the fair online correlation clustering problem can be described by a tuple $\mathcal{I} = (G, a_{1,\ldots,l}, V_{1,\ldots,l}, \sigma)$ where $G = (V, E)$ is the *final graph*, $a_{1,\ldots,l} = \langle a_1, \ldots, a_l \rangle$ contains $l$ positive numbers which represent the fairness parameters, $V_{1,\ldots,l} = \langle V_1, \ldots, V_l \rangle$ contains $l$ subsets of $V$ which represent the colors of the nodes and $\sigma$ is an order on the nodes of $G$: $\sigma = \langle v_1, v_2, \ldots, v_{|V|} \rangle$. For any $0 \leq t \leq |V|$, let $V^t = \langle v_1, v_2, \ldots, v_t \rangle$ be the set of the first $t$ nodes in the order $\sigma$. We refer to these nodes as the nodes that have *arrived until time* $t$, and we refer to $v_t$ as the node arriving at time $t$. Also, we use $n$ to denote the total number of nodes, i.e., $|V| = n$. We let $G_t$ be the subgraph of $G$ induced by $V^t$, $N_{G_t}(u)$ the neighborhood of node $u$ in $G_t$, $V_i^t = V_i \cap V^t$ the nodes with color $i$ arrived until time $t$, $\mathcal{O}_t$ the optimal correlation clustering solution for graph $G_t$ and $\mathcal{O}_t^{fair}$ the optimal fair correlation clustering solution.

**Competitive ratio.** Note that the solution of an algorithm $ALG$ on an online instance $\mathcal{I}$ can be described as a sequence of clustering $\mathcal{C}_1, \mathcal{C}_2, \ldots, \mathcal{C}_{|V|}$. We say that $ALG$ is a $c$-approximation to the optimal fair solution if $\forall t \in \{1, 2, \ldots, |V|\}$ (1) $\text{cost}(\mathcal{C}_t) \leq c \cdot \text{cost}(\mathcal{O}_t^{fair})$; and (2) $\mathcal{C}_t$ is fair with respect to $a$. Additionally, an algorithm is a $c$-competitive algorithm if the solution it produces is a $c$-approximation for all instances $\mathcal{I}$.

**Recourse.** The recourse notion captures how many times a node changes its cluster. To formally define the recourse of a node it is convenient to represent a clustering solution using an *assignment function* $f : V \to \mathbb{Z}$. We say that clustering $\mathcal{C}$ is equivalent to the assignment function $f$ if: (1) $u \sim_{\mathcal{C}} v \Leftrightarrow f(u) = f(v)$; and (2) $u \not\sim_{\mathcal{C}} v \Leftrightarrow f(u) \neq f(v)$. Similarly, we say that a sequence of clustering solutions $\mathcal{C}_1, \mathcal{C}_2, \ldots, \mathcal{C}_{|V|}$ is equivalent to a sequence of assignment functions $f_1, f_2, \ldots, f_{|V|}$ if $f_i$ is equivalent to $\mathcal{C}_i$ for all $i \in \{1, 2, \ldots, |V|\}$ and we denote by $\mathcal{F}_{\mathcal{C}_1, \mathcal{C}_2, \ldots, \mathcal{C}_{|V|}}$ the set of sequences of assignment functions that are equivalent with the clustering sequence $\mathcal{C}_1, \mathcal{C}_2, \ldots, \mathcal{C}_{|V|}$. We define the recourse of a node $u$, arrived at time $t$, with respect to an assignment function sequence $f_1, f_2, \ldots, f_{|V|}$, $r_{f_1, f_2, \ldots, f_{|V|}}(u)$ as the number of times that sequence changes the "cluster id" assigned to $u$. That is $r_{f_1, f_2, \ldots, f_{|V|}}(u) = \sum_{t' > t} \mathbb{I}\{f_{t'-1}(u) \neq f_{t'}(u)\}$. Given a clustering sequence $\mathcal{C}_1, \mathcal{C}_2, \ldots, \mathcal{C}_{|V|}$ the recourse of a node $u$ is defined as $r_{\mathcal{C}_1, \mathcal{C}_2, \ldots, \mathcal{C}_{|V|}}(u) = \inf \left\{ r_{f_1, f_2, \ldots, f_{|V|}}(u) : (f_1, f_2, \ldots, f_{|V|}) \in \mathcal{F}_{\mathcal{C}_1, \mathcal{C}_2, \ldots, \mathcal{C}_{|V|}} \right\}$. The recourse of an algorithm is the worst-case recourse over all instances $\mathcal{I}$ and nodes $u$.

## 3. The Algorithm

In this section, we describe our main algorithm that simultaneously achieves three competing goals: a strong competitive ratio, a low recourse, and a desirable fairness guarantee. Our main technical contribution is to achieve constant-fairness while maintaining the constant-competitiveness and $O(\log n)$ recourse guarantees from Cohen-Addad et al. (2022). In Subsection 3.1 we describe the AGREEMENT subroutine from Cohen-Addad et al. (2021), on which we rely for our algorithm. In

Subsection 3.2, we outline the skeleton of our algorithm, which is structured around three key subroutines. These subroutines, described in Subsection 3.3—MakeFair, Agreement, and Make-Consistent—are responsible for ensuring fairness, maintaining the approximation ratio, and providing recourse guarantees, respectively. The main challenge is for the three subroutines to work together such that the guarantee achieved by one subroutine is not compromised by another subroutine aiming to achieve a different guarantee.

### 3.1. The Agreement Algorithm

Agreement is the algorithm from Cohen-Addad et al. (2021) that takes as input the current graph $G_t$, a parameter $\epsilon \in [0, 1]$, and outputs a clustering $\mathcal{C}^{agr}$. We refer to this clustering as the *agreement decomposition* and rely on it as the base for our other two subroutines for two reasons: (1) its cost is a constant-factor approximation to the optimal solution, and (2) it produces dense clusters which are relatively stable between consecutive iterations. In the rest of the paper, we assume that the input graph contains self-loops, in other words, for any node $u$ we have $u \in N_G(u)$. We start defining two central notions that quantify the similarity between the neighborhood of two nodes and are used by the Agreement algorithm.

**Definition 1 (Agreement)** *Two nodes $u, v$ are in $\epsilon$-agreement in $G$ if*

$$|N_G(u) \triangle N_G(v)| < \epsilon \max\{|N_G(u)|, |N_G(v)|\}$$

*where $\triangle$ denotes the symmetric difference of two sets.*

**Definition 2 (Heaviness)** *A node is called $\epsilon$-heavy if it is in $\epsilon$-agreement with more than a $(1 - \epsilon)$-fraction of its neighbors. Otherwise, it is called $\epsilon$-light.*

The Agreement algorithm uses the agreement and heaviness definitions to compute a solution to the correlation clustering problem. We call the output of Agreement$(G, \epsilon)$ the agreement decomposition of graph $G$. As mentioned before, the cost of this decomposition is a constant factor approximation to the cost of the optimal correlation clustering:

**Lemma 3 (rephrased from Cohen-Addad et al. (2021))** *Let $\mathcal{C}^{agr}$ be the agreement decomposition derived from Agreement$(G, \epsilon)$. There exists constant $\tilde{\epsilon}$ such that for any $\epsilon < \tilde{\epsilon}$ we have that* $\mathrm{cost}(\mathcal{C}^{agr}) \leq \Theta(1)\,\mathrm{cost}(\mathcal{O})$.

The agreement decomposition also exhibits several useful structural properties. Specifically, every non-singleton cluster $C \in \mathcal{C}^{agr}$ is "almost" a clique in the initial graph. We defer the formal statement of these structural properties to Appendix A.

---

**Algorithm 1** The Agreement algorithm

---

**Input:** graph $G$, agreement parameter $\epsilon \in [0, 1]$
Create a graph $G^{filtered}$ from $G$ by discarding all edges whose endpoints are not in $\epsilon$-agreement.
Discard all edges of $G^{filtered}$ between light nodes of $G$.
Compute the connected components of $G^{filtered}$ and output them as the solution.

---

## 3.2. The Algorithm Skeleton

We describe our algorithm, called FAIR-CONSISTENT-AGREEMENT and formally defined in Algorithm 2, as a function of the three subroutines. At each time step $t$, it first computes an initial clustering $\mathcal{C}_t^{agr}$ of the current graph $G_t$ with the AGREEMENT subroutine. Then, for each non-singleton cluster $C$ in $\mathcal{C}_t^{agr}$, it applies the MAKECONSISTENT subroutine to $C$ and obtains a cluster $\widetilde{C} \subseteq C$. It then applies MAKEFAIR to $\widetilde{C}$ and obtains a cluster $C^{\text{fair}} \subseteq \widetilde{C}$. The final clustering $\mathcal{C}_t$ for time step $t$ consists of the clusters $C^{\text{fair}}$ obtained with the non-singleton cluster $C \in \mathcal{C}_t^{agr}$ and singleton clusters for the remaining nodes. The algorithm also initializes and updates variables $g_v$ and $\tilde{t}_v$ for each node $v$, which we discuss in Subsection 3.2 when describing the MAKECONSISTENT subroutine. The main interest is in the three subroutines, which we describe next, and their analyses.

---

**Algorithm 2** The FAIR-CONSISTENT-AGREEMENT algorithm

**Input:** instance $\mathcal{I}$, agreement parameter $\epsilon \in [0,1]$, fairness parameter $\rho \in [0,1]$

**for** each time step $t$ **do**

    Let $v_t$ be the newly arrived node and $G_t$ the updated graph

    Initialize $\mathcal{C}_t \leftarrow \emptyset$, $g_{v_t} \leftarrow 1$, $\tilde{t}_{v_t} \leftarrow t$

    $\mathcal{C}_t^{agr} \leftarrow \text{AGREEMENT}(G_t, \epsilon)$

    **for** each singleton cluster $\{v\} \in \mathcal{C}_t^{agr}$ **do**

        Add $\{v\}$ to $\mathcal{C}_t$

        Update $g_v \leftarrow 0$, $\tilde{t}_v \leftarrow t$

    **for** each non-singleton cluster $C \in \mathcal{C}_t^{agr}$ such that $|C| > 1$ **do**

        $\widetilde{C} \leftarrow \text{MAKECONSISTENT}\big(G_t, \{g_v\}_{v\in C}, \{\tilde{t}_v\}_{v\in C}, C\big)$          $\triangleright \widetilde{C} \subseteq C$

        $C^{fair} \leftarrow \text{MAKEFAIR}\big(G_t, \rho, \widetilde{C}, \mathcal{C}_{t-1}\big)$          $\triangleright C^{fair} \subseteq \widetilde{C}$

        Add $C^{fair}$ and $\{u\}$, for all $u \in C \setminus C^{fair}$, to $\mathcal{C}_t$

    **Output** $\mathcal{C}_t$

---

## 3.3. The Subroutines

**MAKEFAIR**, described in Algorithm 3, takes as input the current graph $G_t$, the fairness tolerance parameter $\rho$, the cluster $\widetilde{C}$ produced by the first subroutine, and $\mathcal{C}_{t-1}$, the clustering constructed by our main algorithm at time $t - 1$. The output is either the entire cluster $\widetilde{C}$ or an empty set, which signifies that the cluster $\widetilde{C}$ is split into singleton clusters in $\mathcal{C}_t$. To determine whether to split $\widetilde{C}$ into singletons, the subroutine calculates the parameter $\rho_t(\widetilde{C})$ and identifies the cluster $C_{prev}$, which is the cluster in $\mathcal{C}_{t-1}$ with the maximum intersection with $\widetilde{C}$. The parameter $\rho_t(\widetilde{C})$ indicates the extent to which $\widetilde{C}$ violates the fairness constraints. The decision rule is as follows: split $\widetilde{C}$ into singletons if $\rho_t(\widetilde{C}) > \rho$; keep $\widetilde{C}$ together if $\rho_t(\widetilde{C}) < \rho/2$. If $\rho_t(\widetilde{C})$ lies within the range $(\rho/2, \rho)$, the subroutine takes a "conservative" approach, repeating the action taken at time $t - 1$. If $|C_{prev}| > 1$, this suggests that $\widetilde{C}$ is part of an evolving cluster that was previously maintained as a single entity. In this case, the subroutine follows the prior decision and outputs $\widetilde{C}$ as it is. In summary, for a cluster $\widetilde{C}$, the subroutine: (1) keeps it together if it is almost fair, i.e., $\rho_t(\widetilde{C}) \leq \rho/2$; (2) splits it if it is

very unfair, i.e., $\rho_t(\widetilde{C}) \geq \rho$; and (3) repeats the previous decision if it is neither almost fair nor very unfair. The reason for introducing a multiplicative gap between what is considered almost fair and very unfair is to ensure that the evolving cluster must increase multiplicatively in size each time the decision on whether to split it or not changes. The latter helps us bound the increase in recourse incurred due to maintaining fair clusters.

---

**Algorithm 3** The MAKEFAIR subroutine

---

    **Input:** current graph $G_t$, fairness parameter $\rho \in [0, 1]$, cluster $\widetilde{C}$, and clustering $\mathcal{C}_{t-1}$

    **If** $|\widetilde{C}| = 1$ **then return** $\widetilde{C}$

    $\rho_t(\widetilde{C}) \leftarrow \max\{r : |V_i \cap \widetilde{C}| \leq (1 + r)a_i|\widetilde{C}| \quad \forall i\}$

    $C_{prev} \leftarrow \operatorname{argmax}\{|\widetilde{C} \cap C'| : C' \in \mathcal{C}_{t-1}\}$

    /* Small fairness violation or medium violation and cluster existed at time $t - 1$ */

    **If** $\rho(\widetilde{C}) \leq \rho/2$ or $(\rho_t(\widetilde{C}) \in (\rho/2, \rho)$ and $|C_{prev}| > 1\})$ **then return** $\widetilde{C}$

    /* Large violation of fairness or cluster did not exist at time $t - 1$, split cluster */

    **Else return** $\emptyset$

---

**MAKECONSISTENT** takes as input the current graph $G_t$, two labeling functions $g$ and $\tilde{t}$, and a non-singleton cluster $C$ from the current agreement decomposition. The subroutine outputs a cluster $\widetilde{C} \subseteq C$, where the nodes in $C \setminus \widetilde{C}$ will be clustered as singletons in the final solution $\mathcal{C}_t$, while cluster $\widetilde{C}$ may either be entirely split into singletons by MAKEFAIR or preserved as is. The subroutine heavily relies on the two labeling functions $g$ and $\tilde{t}$, which store important information regarding previous clustering decisions for each node $u \in C$. As a first step, the subroutine checks whether the set $\{u \in: g_u = 0\}$ is too large-—specifically, if its size exceeds $100\epsilon|C|$—the algorithm sets $g_u = 0$ for all nodes in $C$. Next, among the nodes with $g_u = 0$, we compute the one whose degree was minimum at time $\tilde{t}_u$. If that node, denoted as $u_C, t$, has increased its degree by a factor of at least $(1 + 100\epsilon)$, we update the labeling functions $g$ and $\tilde{t}$ for all nodes in $C$ and output $\widetilde{C} = C$. If this condition is not met, we output $\widetilde{C}$ and cluster all nodes with $g_u = 0$ as singletons. The function $g$ is designed to track whether a node was clustered as a singleton in the previous iteration. Specifically, if $g_u = 0$ at the beginning of iteration $t$, then node $u$ was clustered as a singleton at time $t - 1$, and $g_u$ is updated accordingly at each iteration. The variable $\tilde{t}_u$ is initialized to the arrival time of node $u$ and is updated to the current time $t$ under two conditions: (1) when $u$ is clustered as a singleton by the agreement decomposition (see FAIR-CONSISTENT-AGREEMENT), or (2) when $g_u = 0$ and the condition in the second "if" statement of MAKECONSISTENT is satisfied. Thus, $\tilde{t}_u$ records the most recent occurrence of one of the following events: (1) $u$'s arrival; (2) the last time $u$ was clustered as a singleton (if it has ever been); or (3) the last time $u$ was part of a cluster where the condition in the second "if" statement was met (if this has ever occurred).

MAKECONSISTENT seeks to maintain a constant-factor approximate clustering while minimizing the recourse. Since $\mathcal{C}_t^{agr}$ is already a constant approximate solution, we aim to set $\widetilde{C} \simeq C$ whenever feasible. However, to reduce recourse, we only "add" nodes previously clustered as singletons if their degree has increased by a multiplicative factor. That way, if $\widetilde{C} = C$, any recourse increase for nodes with $g_u = 0$ is attributed to their degree growth. Otherwise, the cost of clustering them as singletons is charged to previous agreement decompositions.

---

**Algorithm 4** The MAKECONSISTENT subroutine

---

 **Input:** graph $G_t$, cluster $C$, node labelings $\{g_v\}_{v \in C}$ and $\{\tilde{t}_v\}_{v \in C}$

 **if** $|\{u \in C : g_u = 0\}| > 100\epsilon|C|$ **then**

  $g_v \leftarrow 0, \forall v \in C$

 $u_{C,t} \leftarrow \mathrm{argmin}_{u \in C : g_u = 0} |N_{G_{\tilde{t}_u}}(u)|$

 **if** $|N_{G_t}(u_{C,t})| > (1 + 20\epsilon)|N_{G_{\tilde{t}_{u_{C,t}}}}(u_{C,t})|$ **then**

  $g_v \leftarrow 1, \tilde{t}_v \leftarrow t, \forall v \in C$

  **return** $C$

 **else**

  **return** $C \setminus \{u \in C : g_u = 0\}$

---

## 4. Analysis of the Recourse

In this section, we prove that if $\epsilon$ is set to a small enough constant then the worst-case recourse of a node is $O(\log n)$, formally stated in Theorem 1. In the remainder of the paper, we assume that all nodes $u$ have degree $|N(u)| \geq \frac{1}{2\epsilon}$. We show that this is without loss of generality in Appendix E.

**Theorem 1** *There exists a small enough constant $\tilde{\epsilon}$ such that for any $\epsilon < \min\{\frac{\rho \min_i a_i}{11200}, \tilde{\epsilon}\}$, the recourse of Algorithm 2 is $O\left(\frac{\log n}{\epsilon}\right)$.*

To prove Theorem 1 we upper bound the number of times that a node transitions from being a singleton to a non-singleton cluster in our solution. In Lemma 7 of the appendix we prove that the latter quantity is at most two times the recourse, consequently, upper bounding it implies the same (up to constant terms) upper bound for the recourse. Formally, let $\mathcal{S}(\mathcal{C})$ denote the singleton clusters of clustering $\mathcal{C}$, then we define $r^{aux}_{\mathcal{C}_1, \mathcal{C}_2, \ldots, \mathcal{C}_{|V|}}(u) = \sum_{t' > t} \mathbb{I}\{u \in S(\mathcal{C}_{t'}) \setminus S(\mathcal{C}_{t'-1})\}$, prove (in Lemma 7) that Algorithm 2 produces clustering sequences where $r_{\mathcal{C}_1, \mathcal{C}_2, \ldots, \mathcal{C}_{|V|}}(u) \leq 2 \cdot r^{aux}_{\mathcal{C}_1, \mathcal{C}_2, \ldots, \mathcal{C}_{|V|}}(u) + 1$, and focus in the rest of the section on proving that $r^{aux}_{\mathcal{C}_1, \mathcal{C}_2, \ldots, \mathcal{C}_{|V|}}(u) \leq O(\log n)$. To ease notation, whenever the clustering sequence that we are referring to is clear from the context we write $r^{aux}(u)$ instead of $r^{aux}_{\mathcal{C}_1, \mathcal{C}_2, \ldots, \mathcal{C}_{|V|}}(u)$. Also, to better describe increases of $r^{aux}(u)$ we define the partial sums function $r^{aux}_{t_A, t_B}(u) = \sum_{t \in [t_A + 1, t_B]} \mathbb{I}\{u \in S(\mathcal{C}_{t-1}) \setminus S(\mathcal{C}_t)\}$ and say that "$r^{aux}(u)$ increases at time $t$" if $\mathbb{I}\{u \in S(\mathcal{C}_t) \setminus S(\mathcal{C}_{t-1})\} = 1$. The proof proceeds by arguing that a constant increase of $r^{aux}(u)$ between times $t'$ and $t$ implies a multiplicative increase of $u$'s degree between the same times $t'$ and $t$. Thus, since the degree of any node is upper bounded by $n$ we deduce that $r^{aux}(u)$ can increase at most $O(\log n)$ times.

As aforementioned in Section 3.3, in the intermediate clustering computed by MAKECONSISTENT, $g_u$ is clustered in a non-singleton cluster if and only if $g_u = 1$. Thus, whenever the label of node $u$ transitions from 0 to 1 its recourse increases. A crucial concept in our analysis, which will help us bound $r^{aux}(u)$, is the *Important event* definition.

**Definition 1** *Let $C$ be a non-singleton cluster of $\mathcal{C}_t^{agr}$ and node $u_{C,t}$ be as calculated by Algorithm 4 at time $t$. If $|N_{G_t}(u_{C,t})| > (1 + 20\epsilon)|N_{G_{\tilde{t}_{u_{C,t}}}}(u_{C,t})|$ then we say that cluster $C$ and all its nodes participate in an important event at that time.*

Definition 1 tries to capture transitions of $g_u$ from 0 to 1. Those transitions are related to the recourse increase due to the online nature of a problem. In Lemma 2 we relate those transitions, through Definition 1, with the degree increase of a node.

**Lemma 2** *Let $t_1 < t_2$ be two times that $u$ participates in an important event. Then $|N_{G_{t_2}}(u)| \geq (1 + \epsilon)|N_{G_{t_1}}(u)|$*

However, the recourse can also increase between times where the label $g_u$ remains 1 due to the MAKEFAIR procedure which keeps the output fair. We deal with such increases in Lemma 3.

**Lemma 3** *Let $t_A < t_D$ be two consecutive times that node $u$ participates in an important event and let $t_B, t_C \in [t_A, t_D]$ be such that $r_{t_B,t_C}^{aux}(u) > 1$. If $\epsilon$ is a small enough constant for which $\epsilon < \frac{a_i \rho}{100 \cdot 112}$ then $|N_{G_{t_C}}(u)| \geq (1 + \epsilon)|N_{G_{t_B}}(u)|$.*

We continue by proving the main Theorem 1 and then provide the more technical proof of Lemma 3 while we defer the proof of Lemma 2 to Appendix B.

**Proof of Theorem 1.** Fix node $u$, from Lemma 7 we know that $r(u) \leq 2 \cdot r^{aux}(u) + 1$ therefore it suffices to upper bound $r^{aux}(u)$. To that end, we show that between the times when $r^{aux}(u)$ increases by a constant amount, the degree of $u$ increases multiplicatively. Then the statement follows as the degree is at most the number of nodes $n$. Now suppose that $t_1 < t_2 < \cdots < t_k$ are the times when $u$ participates in an important event. Note that at these times $r^{aux}(u)$ can never increase as it is then that $u$ transitions in the opposite way, from singleton to clustered. Now let $r$ be the times $r^{aux}(u)$ increases between two consecutive important events at $t_i$ and $t_{i+1}$:

- $r = 0$: It is irrelevant whether the degree of $u$ increases.

- $r = 1$: In this case Lemma 2 suffices to show that the degree of $u$ increased multiplicatively by a factor of $1 + \epsilon$.

- $r \geq 2$: We can partition the interval $[t_i, t_{i+1}]$ into $r/2$ consecutive intervals so that in each of them there are exactly two increases of $r^{aux}(u)$. Then applying Lemma 3 to each of them gives a total increase of $(1 + \epsilon)^{r/2}$ to the degree of $u$.

Overall, for every two consecutive increases of $r^{aux}(u)$ the degree of $u$ increases at least by $1 + \epsilon$. Then we have,

$$r^{aux}(u) \leq 2 \cdot \log_{1+\epsilon} n = 2 \cdot \frac{\log n}{\log(1 + \epsilon)} \leq 2 \cdot \left(1 + \frac{1}{\epsilon}\right) \log n = O\left(\frac{\log n}{\epsilon}\right)$$

where the second inequality is due to $\log(1 + \epsilon) \geq \frac{\epsilon}{1+\epsilon}$ since $\epsilon > -1$. ∎

**Proof of Lemma 3.** Let $\mathcal{C}_t^{agr}, \widetilde{C}_t, C_t$ be the clusters to which $u$ is clustered by AGREEMENT, the intermediate clustering produced by MAKECONSISTENT and the final cluster produced by FAIR-CONSISTENT-AGREEMENT. Since $r_{t_B,t_C}^{aux}(u) > 1$ there exist $t_1 < t_2 < t_3 \in (t_A, t_B)$ such that $|C_{t_1}| > 1, |C_{t_2}| = 1$ and $|C_{t_3}| > 1$ and it is enough to prove that $|N_{G_{t_3}}(u)| \geq (1 + \epsilon)|N_{G_{t_1}}(u)|$. Without loss of generality, we assume that among all such triplets, $t_3$ is chosen to be the smallest, and $t_2$ is the smallest time in $(t_1, t_3)$ such that $|C_{t_2}| = 1$. Note that this implies $|C_{t_2-1}| > 1$ and $|C_{t_3-1}| = 1$.

First note that $|\mathcal{C}_t^{agr}| > 1$ and $|\widetilde{C}_t| > 1$ for all $t \in [t_A, t_3]$. Indeed, if the latter was not true then we would have $|C_{t'}| = 1$ for some $t' \in [t_A, t_3]$. Consequently, at that time $t'$ we would have set $g_u = 0$ and node $u$ could be reinserted in a non-singleton cluster only through an "important" event, which happens at time $t_D > t_3$, contradicting that $|C_3| > 1$. Similarly, we can argue that $\forall t \in [t_A, t_3] \; |u \in \mathcal{C}_t^{agr} : g_u = 0| \leq 100\epsilon|\mathcal{C}_t^{agr}|$, otherwise $g_u$ would have been set to 0 at some $t' \in [t_a, t_3]$ which contradicts $|C_3| > 1$. Thus for all times $t \in [t_1, t_3]$ we have that:

$$|\mathcal{C}_t^{agr} \setminus \widetilde{C}_t| \leq 100\epsilon|\mathcal{C}_t^{agr}| \Leftrightarrow |\widetilde{C}_t| \geq (1 - 100\epsilon)|\mathcal{C}_t^{agr}| \qquad (\star)$$

and:

$$\begin{aligned} |N_{G_t}(u) \cap \widetilde{C}_t| &\geq |N_{G_t}(u) \cap \mathcal{C}_t^{agr}| - |\mathcal{C}_t^{agr} \setminus \widetilde{C}_t| \\ &\geq (1 - 9\epsilon)|\mathcal{C}_t^{agr}| - 100\epsilon|\mathcal{C}_t^{agr}| \\ &= (1 - 109\epsilon)|\mathcal{C}_t^{agr}| \end{aligned}$$

where in the second inequality we used Eq. $(\star)$ and Property 4. Combining this last inequality with $|\mathcal{C}_t^{agr}| \geq (1 - 3\epsilon)|N_{G_t}(u)|$ from Property 1 and the fact that $\mathcal{C}_t^{agr} \supseteq \widetilde{C}_t$. We get:

$$\begin{aligned} |N_{G_t}(u) \cap \widetilde{C}_t| &\geq \max\left\{(1 - 109\epsilon)|\widetilde{C}_t|, (1 - 109\epsilon)(1 - 3\epsilon)|N_{G_t}(u)|\right\} \\ &\geq (1 - 112\epsilon)\max\left\{|\widetilde{C}_t|, |N_{G_t}(u)|\right\}. \end{aligned}$$

From this last inequality we have that $N_{G_{t_2}}(u) \simeq \widetilde{C}_{t_2}$ and $N_{G_{t_3}}(u) \simeq \widetilde{C}_{t_3}$. At the same time since $|C_{t_2-1}| > 1$, $|\widetilde{C}_{t_2}| > 1$ and $|C_{t_2}| = 1$ we deduce that $\widetilde{C}_{t_2}$ is not $\rho$-fair (otherwise it would not have been split into singletons by MAKEFAIR). Similarly, since $|C_{t_3-1}| = 1$, $|\widetilde{C}_{t_3-1}| > 1$ and $|C_{t_3}| > 1$ we deduce that $\widetilde{C}_{t_3}$ is $\rho/2$-fair. Consequently, there exists $i \in \{1, \dots, l\}$ such that $|\widetilde{C}_{t_2} \cap V_i| > (1 + \rho)a_i|\widetilde{C}_{t_2}|$ and $|\widetilde{C}_{t_3} \cap V_i| \leq (1 + \rho/2)a_i|\widetilde{C}_{t_3}|$. Next,

$$\begin{aligned} |N_{G_{t_2}}(u) \cap V_i| &\geq |\widetilde{C}_{t_2} \cap V_i| - |N_{G_{t_2}}(u) \setminus \widetilde{C}_{t_2}| \\ &\geq (1 + \rho)a_i|\widetilde{C}_{t_2}| - 112\epsilon|\widetilde{C}_{t_2}| \\ &= (a_i + a_i\rho - 112\epsilon)|\widetilde{C}_{t_2}| \\ &= (a_i + a_i\rho - 112\epsilon)(1 - 112\epsilon)|N_{G_{t_2}}(u)|. \end{aligned}$$

Likewise we upper bound $|N_{G_{t_3}}(u) \cap V_i|$ as follows:

$$\begin{aligned} |N_{G_{t_3}}(u) \cap V_i| &\leq |\widetilde{C}_{t_3} \cap V_i| + |N_{G_{t_3}}(u) \setminus \widetilde{C}_{t_3}| \\ &\leq (1 + \rho/2)a_i|\widetilde{C}_{t_3}| + 112\epsilon|\widetilde{C}_{t_3}| \\ &= (a_i + (\rho/2)a_i + 112\epsilon)\frac{N_{G_{t_3}}(u)}{1 - 112\epsilon}. \end{aligned}$$

Combining the upper bound on $|N_{G_{t_3}}(u) \cap V_i|$, the lower bound on $|N_{G_{t_2}}(u) \cap V_i|$ and the fact that $N_{G_{t_2}}(u) \subseteq N_{G_{t_3}}(u) \Rightarrow |N_{G_{t_2}}(u) \cap V_i| \leq |N_{G_{t_3}}(u) \cap V_i|$ we have:

$$(a_i + \rho/2 a_i + 112\epsilon) \frac{N_{G_{t_3}}(u)}{1 - 122\epsilon} \geq (a_i + a_i\rho - 112\epsilon)(1 - 112\epsilon)|N_{G_{t_2}}(u)| \Rightarrow \tag{1}$$

$$\frac{N_{G_{t_3}}(u)}{N_{G_{t_2}}(u)} \geq \frac{(a_i + a_i\rho - 112\epsilon)(1 - 112\epsilon)^2}{(a_i + \rho/2 a_i + 112\epsilon)} \tag{2}$$

$$\geq \frac{(1 + \rho - 112\epsilon/a_i)(1 - 112\epsilon)^2}{(1 + \rho/2 + 112\epsilon/a_i)} \tag{3}$$

$$\geq \frac{(1 + 99\rho/100)(1 - 112\epsilon)^2}{(1 + 51\rho/100)} \tag{4}$$

$$\geq (1 + \rho/10)(1 - 112\epsilon)^2 \tag{5}$$

$$\geq (1 + 1000\epsilon)(1 - 112\epsilon)^2 \tag{6}$$

$$\geq (1 + \epsilon) \tag{7}$$

where from (3) to (4) we used the left-hand side and in (5) to (6) the right-hand side of the inequality $\epsilon < \frac{a_i\rho}{100 \cdot 112} < \frac{\rho}{10000}$. ∎

## 5. Analysis of the Competitive Ratio

In this section, we show that FAIR-CONSISTENT-AGREEMENT (Algorithm 2) is a constant-factor approximation to the optimal fair clustering at every step. Note that at each time $t$, our algorithm may only modify non-singleton clusters of $\mathcal{C}_t^{agr}$. Consequently, only these modifications may increase the cost it incurs with respect to the agreement decomposition, and the goal of this section is to bound such increases.

**Theorem 4**  *There exists a small enough constant $\tilde{\epsilon}$ such that for any $\epsilon < \min\{\frac{\rho \min_i a_i}{11200}, \tilde{\epsilon}\}$, we have that $\mathrm{cost}(\mathcal{C}_t) \leq \Theta\left(\frac{1}{\epsilon^3 \rho \min_i a_i^2}\right) \cdot \mathrm{cost}(\mathcal{O}_t^{fair})$ for any $t$.*

Given a clustering $\mathcal{C}$, let $\mathcal{C}^{\mathrm{fair}}$ be a clustering identical to $\mathcal{C}$, except that any non $\rho$-fair cluster $C \in \mathcal{C}$ is split into singletons. In Theorem 4 we argue that the cost of $\mathcal{C}^{\mathrm{fair}}$ is a constant-factor approximation to the cost of the optimal fair solution. In other words, splitting very unfair clusters of a good solution does not substantially affect our approximation guarantee. In addition, let $\widetilde{\mathcal{C}}$ be the clustering produced by running MAKECONSISTENT across all $C \in \mathcal{C}^{agr}$ (we formally define this clustering in Appendix C), and note that the clustering which our algorithm outputs is essentially $\widetilde{\mathcal{C}}^{fair}$. Given Lemma 6 to conclude the main Theorem 4 it is enough to argue that $\mathrm{cost}(\widetilde{\mathcal{C}}_t) \leq \Theta(1) \cdot \mathrm{cost}(\mathcal{O}_t)$ which we do in Lemma 5 of Appendix C.

**Lemma 5**  *For a constant $\epsilon$ small enough we have $\mathrm{cost}(\widetilde{\mathcal{C}}_t) \leq \Theta\left(\frac{1}{\epsilon^3}\right) \cdot \mathrm{cost}(\mathcal{O}_t)$ for any $t$.*

For the rest of the section, we focus on proving Lemma 6.

**Lemma 6**  *Let $\widetilde{\mathcal{C}}$ be the output of any constant-factor approximation algorithm to the unconstrained problem and $\mathcal{C}^{fair}$ be the clustering which results in splitting all non $\rho/2$-fair clusters of $\widehat{\mathcal{C}}$ into singletons. Then:*

$$\mathrm{cost}(\mathcal{C}^{fair}) \leq \Theta\left(\frac{1}{\rho \min_i a_i^2}\right) \cdot \mathrm{cost}(\mathcal{O}^{fair}).$$

**Proof** Let $\widetilde{\mathcal{C}}_{\text{split}} \subseteq \widetilde{\mathcal{C}}$ be the set of clusters where the relaxed fairness condition is violated, we have that $\text{cost}(\mathcal{C}^{fair}) \leq \text{cost}(\widetilde{\mathcal{C}}) + \sum_{C \in \widetilde{\mathcal{C}}_{\text{split}}} |C|^2$. We argue that for every $C \in \widetilde{\mathcal{C}}_{\text{split}}$ either the optimal fair clustering or the optimal clustering to the unconstrained problem "pays" $\simeq \Theta\left(\frac{1}{\rho \min_i a_i^2}\right) |C|^2$. We do that by identifying two large sets of nodes in each case, for which all in-between edges (which we call $E_C^+$) and non-edges (which we will call $E_C^-$) contribute to these. We note that each of the charged edges or non-edges will be charged by only one cluster $C$. Let $C$ be one of those clusters and let $i$ be the color for which the relaxed fair constraint is violated, then we know that $|C \cap V_i| > (1 + \rho/2)a_i|C|$ as the algorithm only splits a cluster with a medium or large violation. Now, denote by $C' \in \mathcal{O}^{fair}$ the cluster in the optimal fair solution which contains the maximum number of nodes in $C \cap V_i$. We distinguish between two cases:

1. $|C' \cap C \cap V_i| > (1 + \rho/4)a_i|C|$ in which case we charge $|C|^2$ to the edges between $C'$ and $C$; and

2. $|C' \cap C \cap V_i| \leq (1 + \rho/4)a_i|C|$ in which case we charge $|C|^2$ to the edges between nodes in $C \cap V_i$ that belong to different clusters of $\mathcal{O}_{fair}$.

In the first case using the condition above as well as the fact that $C' \in \mathcal{O}^{fair}$ we have:

$$(1 + \rho/4)a_i|C| \leq |C' \cap C \cap V_i| \leq |C' \cap V_i| \leq a_i|C'| \tag{$*$}$$

Then we can prove that $C'$ has a lot of elements outside of $C$:

$$|C' \setminus C| \geq |C'| - |C| \geq \rho/4|C|.$$

where the second inequality comes from Equation ($*$) after dividing by $a_i$.

The latter implies that in $C'$ there are two "large" sets of nodes, i.e., $C' \setminus C$ and $C' \cap C \cap V_i$ that were not clustered together in $C$. Between these sets there are many tuples $\{(u, v) : u \in C' \setminus C, v \in C' \cap C \cap V_i\}$ which we partition in $E_C^+$ and $E_C^-$ for edges and non-edges respectively. Then the edges of $E_C^+$ are cut by $C$ and contribute to $\widetilde{\mathcal{C}}$ while the non-edges of $E_C^-$ are paid by $C'$ and contribute to $\mathcal{O}^{fair}$. Therefore every one of these tuples can be charged to $\text{cost}(\mathcal{O}^{fair}) + \text{cost}(\widetilde{\mathcal{C}})$ and there are many of them:

$$|E_C^+| + |E_C^-| = |C' \setminus C| \cdot |C' \cap C \cap V_i| \geq \rho/4|C| \cdot (1 + \rho/4)a_i|C| \geq \Theta(\rho) \min_i a_i|C|^2$$

Then we can upper bound $|C|^2$:

$$|C|^2 \leq \Theta\left(\frac{1}{\rho \min_i a_i}\right)\left(|E_C^+| + |E_C^-|\right). \tag{1}$$

In the second case, we know that $|C' \cap C \cap V_i|$ is small for any $C' \in \mathcal{O}^{fair}$, therefore $\mathcal{O}^{fair}$ leaves a lot of tuples between its clusters within $C \cap V_i$. We proceed by constructing a set of nodes $S$ using clusters of $\mathcal{O}^{fair}$ so that there are a lot of tuples between $|S \cap C \cap V_i|$ and $|(C \cap V_i) \setminus S|$. The set $S$ satisfies the property:

$$\frac{1}{2}(1 + \rho/4)a_i|C| \leq |S \cap C \cap V_i| \leq (1 + \rho/4)a_i|C|.$$

We start by ordering $C'_j \in \mathcal{O}^{fair}$ by decreasing $|C'_j \cap C \cap V_i|$. Then we greedily add as many of them as we can to $S$ so that the upper bound of the property is not violated. Suppose that we have added $k$ clusters and adding the next one violates the property. Notice that by definition $|C'_{k+1} \cap C \cap V_i| \leq |C'_i \cap C \cap V_i|$ for $i \leq k$. By summing we have:

$$k \cdot |C'_{k+1} \cap C \cap V_i| \leq \sum_{i=1}^{k} |C'_i \cap C \cap V_i| = |S \cap C \cap V_i|.$$

Since adding the cluster $C'_{k+1}$ violates the property and using the last inequality we have:

$$
\begin{aligned}
(1 + \rho/4) a_i |C| &< |(S \cup C'_{k+1}) \cap C \cap V_i| \\
&= |S \cap C \cap V_i| + |C'_{k+1} \cap C \cap V_i| \\
&\leq \left(1 + \frac{1}{k}\right) \cdot |S \cap C \cap V_i|
\end{aligned}
$$

where the equality is due to $C'_{k+1}$ being a different cluster, therefore having no intersection with $S$. Using $k \geq 1$ we obtain the lower bound of the property. Now we can proceed to show that there are many remaining elements of $|C \cap V_i|$ outside of $S$. Using the fact that $C$ violates fairness by $\rho/2$ for $V_i$ we have $|C \cap V_i| \geq (1 + \rho/2) a_i |C|$. Then the nodes of $C \cap V_i$ not in $S$ are:

$$|(C \cap V_i) \setminus S| = |C \cap V_i)| - |C \cap V_i \cap S| \geq (1 + \rho/2) a_i |C| - (1 + \rho/4) a_i |C| = (\rho/4) a_i |C|$$

Between that and the rest nodes of $C \cap V_i$ there are many tuples $\{(u, v) : u \in S \cap C \cap V_i, v \in (C \cap V_i) \setminus S\}$, of which those that are edges ($E_C^+$ as before) contribute to $\mathcal{O}^{fair}$ since they belong to different clusters, while those that are non-edges ($E_C^-$ as before) contribute to $\widetilde{\mathcal{C}}$ as both nodes are in $C$ and there are many of them:

$$|E_C^+| + |E_C^-| = |S \cap C \cap V_i| \cdot |(C \cap V_i) \setminus S| \geq \frac{1}{2}(1 + \rho/4) a_i |C| \cdot (\rho/4) a_i |C| \geq \Theta(\rho) \min_i a_i^2 |C|^2.$$

Then we can upper bound $|C|^2$:

$$|C|^2 \leq \Theta\left(\frac{1}{\rho \min_i a_i^2}\right) \left(|E_C^+| + |E_C^-|\right). \tag{2}$$

Overall we can see that in both cases

$$
\begin{aligned}
\text{cost}(\mathcal{C}^{fair}) &\leq \text{cost}(\widetilde{\mathcal{C}}) + \sum_{C \in \widetilde{\mathcal{C}}_{\text{split}}} |C|^2 \leq \text{cost}(\widetilde{\mathcal{C}}) + \Theta\left(\frac{1}{\rho \min_i a_i^2}\right) \sum_{C \in \widetilde{\mathcal{C}}_{\text{split}}} (|E_C^+| + |E_C^-|) \\
&\leq \text{cost}(\widetilde{\mathcal{C}}) + \Theta\left(\frac{1}{\rho \min_i a_i^2}\right) \left(\text{cost}(\widetilde{\mathcal{C}}) + \text{cost}(\mathcal{O}^{fair})\right) \leq \Theta\left(\frac{1}{\rho \min_i a_i^2}\right) \text{cost}(\mathcal{O}^{fair}).
\end{aligned}
$$

where the second inequality is due to Equation (1), Equation (2) as $\frac{1}{\rho \min_i a_i} \leq \frac{1}{\rho \min_i a_i^2}$, the third inequality due to all tuples in $|E_C^+|, |E_C^-|$ contributing to $\text{cost}(\widetilde{\mathcal{C}}) + \text{cost}(\mathcal{O}^{fair})$ (note that the tuples are counted once each across all $C$) and the last inequality due to $\widetilde{\mathcal{C}}$ being a constant approximation of the optimal unconstrained cost which lower bounds the optimal fair cost. ∎

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

## Appendix A. Structural Properties of the *Agreement* Decomposition

Let $\mathcal{C}$ be the clustering produced by AGREEMENT$(G, \epsilon)$ and $u, v$ two nodes which belong to the same cluster $C \in \mathcal{C}$, then for $\epsilon$ small enough the following properties hold, which were shown in Cohen-Addad et al. (2021).

**Property 1** $|N_G(u) \cap C| \geq (1 - 3\epsilon)|N_G(u)|$

**Property 2** $|N_G(u) \setminus C| < 3\epsilon|N_G(u)|$

**Property 3** $|C| \geq (1 - 3\epsilon)|N_G(u)|$

**Property 4** $|N_G(u) \cap C| \geq (1 - 9\epsilon)|C|$

**Property 5** $|C \setminus N_G(u)| < 9\epsilon|C|$

**Property 6** $|N_G(u)| \geq (1 - 9\epsilon)|C|$

**Property 7** $|N_G(u) \cap N_G(v)| \geq (1 - 5\epsilon) \max\{|N_G(u)|, |N_G(v)|\}$

**Property 8** $|N_G(v)|(1 - 5\epsilon) \leq |N_G(u)| \leq \frac{|N_G(v)|}{1 - 5\epsilon}$

**Property 9** $|C \setminus N_G(u)| < 9\epsilon|C| < \frac{9\epsilon}{1 - 9\epsilon}|N_G(u)|$

**Property 10** $|N_G(u) \setminus C| < 3\epsilon|N_G(u)| < \frac{3\epsilon}{1 - 3\epsilon}|C|$

**Property 11** $N_G(u) \cap N_G(v) \neq \emptyset$

## Appendix B. Additional Lemmas for Bounding the Recourse

This section is devoted to giving more intuition on how to bound the recourse. We start by proving that bounding $r^{aux}$ suffices to bound the recourse, then we make a series of observations regarding the inner workings of our algorithm, and finally, we add the proof of the missing Lemma 3.

### B.1. Cluster id Assignment

To bound the recourse we need to specify how the ids are assigned to the clusters with an assignment function $f_t$ that is equivalent to $\mathcal{C}_t$ at every step $t$. Then due to Lemma 9, this function can have the following simple form. If we have a singleton node at time $t$ that was singleton at time $t - 1$ as well, then we use the same id, otherwise, it was split from a cluster and it is assigned a new id. On the other hand, if we have a non-singleton cluster $C$ at time $t$, we check whether this cluster existed at $t - 1$, in which case we use the same id, or if it is a new cluster. In terms of notation it is useful to see the is as a cluster property and for any cluster $C \subseteq V$ define:

$$f_t(C) = y \Leftrightarrow f_t(u) = y \; \forall u \in C.$$

Furthermore, it is useful to define for any clustering $\mathcal{C}$ the set of singleton nodes $\mathcal{S}(\mathcal{C}) = \{u \in C \in \mathcal{C} : |C| = 1\}$ and the set of nodes of non-trivial clusters $\mathcal{R}(\mathcal{C}) = \{u \in C \in \mathcal{C} : |C| > 1\}$.

**Lemma 7** *The recourse of node $u$ is upper bounded by $2 \cdot r^{aux}(u) + 1$.*

**Proof** Suppose that $u$ is a node that appears in the input at time $t$. For its recourse, we have:

$$
\begin{aligned}
r_{\mathcal{C}_1,\ldots,\mathcal{C}_{|V|}}(u) &\leq r_{f_1,\ldots,f_{|V|}}(u) \\
&= \sum_{t' > t} \mathbb{I}\{f_{t'-1}(u) \neq f_{t'}(u)\} \\
&= \sum_{t' > t} \mathbb{I}\{u \in S(\mathcal{C}_{t'}) \setminus S(\mathcal{C}_{t'-1})\} + \sum_{t' > t} \mathbb{I}\{u \in R(\mathcal{C}_{t'}) \setminus R(\mathcal{C}_{t'-1})\} \quad (1)
\end{aligned}
$$

where the first inequality follows from the definition of $r$ given that $f_t$ is equivalent to $\mathcal{C}_t$ due to Lemma 8, the second equality follows from the definition of $f_t$ which only changes its value for $u$

---

**Algorithm 5**

> **function** ID-UPDATE($\mathcal{C}_t, \mathcal{C}_{t-1}, id_{max}$)
>> **for** $C \in \mathcal{C}_t$ **do**
>>> $C_{prev} \leftarrow \arg\max\{|C' \cap C| : C' \in \mathcal{C}_{t-1}\}$
>>> **if** $|C| = 1$ **then**
>>>> **if** $|C_{prev}| = 1$ **then** $f_t(C) \leftarrow f_{t-1}(C)$
>>>> **else** $f_t(C) \leftarrow id_{max}{+}{+}$
>>> **else**
>>>> **if** $|C_{prev}| = 1$ **then** $f_t(C) \leftarrow id_{max}{+}{+}$
>>>> **else** $f_t(C) \leftarrow f_{t-1}(C_{prev})$
>> **return** $f_t, id_{max}$

---

when it transitions from being in a cluster to singleton or vice versa. We proceed to show that the right-hand side of 1 is upper bounded by:

$$2 \cdot \sum_{t' > t} \mathbb{I}\{u \in S(\mathcal{C}_{t'}) \setminus S(\mathcal{C}_{t'-1})\} + 1$$

which suffices for the statement. Define a sequence $t = t_0 \leq t_1 \leq \cdots \leq T$ such that $u$ incurs 1 recourse only on these iterations $t_i$ and $T$ is the last iteration. We consider two cases, based on whether $u$ is singleton or not at iteration $t$. First, assume that it was in a non trivial cluster $C$, then on $t_i$ such that $i$ is even, $u$ is in a non-trivial cluster and singleton for odd $i$. Define $i^* = \arg\max\{i : t_i \leq T\}$. If $i$ is even, we have that $r_{\mathcal{C}_1,...,\mathcal{C}_{|V|}}(u) = 2 \cdot \sum_{t' > t} \mathbb{I}\{u \in S(\mathcal{C}_{t'}) \setminus S(\mathcal{C}_{t'-1})\}$, otherwise $r_{\mathcal{C}_1,...,\mathcal{C}_{|V|}}(u) = 2 \cdot \sum_{t' > t} \mathbb{I}\{u \in S(\mathcal{C}_{t'}) \setminus S(\mathcal{C}_{t'-1})\} - 1$ and in both cases the statement holds. Second, assume that on iteration $t$, $u$ is singleton. If $t_1 > T$ then $r_{\mathcal{C}_1,...,\mathcal{C}_{|V|}}(u) = 0$, while if $t_1 \leq T$, $u$ incurs 1 recourse and joins a cluster on that iteration. From that point on, our previous analysis on starting from being in a cluster holds, therefore the statement follows. ∎

**Lemma 8** *The function $f_t$ defined in Algorithm 5 is an assignment function and is equivalent to $\mathcal{C}_t$ for any $t$.*

**Proof** We start by showing that $f_t$ is a well-defined assignment function. We can show that it is possible to track the evolution of a cluster. First, note that $\mathcal{C}_t$ only differs from $\mathcal{C}_t^{agr}$ in that a part of some clusters has been split into singletons, therefore the statement of Lemma 9 holds for $\mathcal{C}_t$ as well. Therefore, between $t-1$ and $t$ a cluster $C' \in \mathcal{C}_{t-1}$ can remain the same, have some or all of its nodes split to singletons, or gain some singletons, but it can never receive nodes from another cluster as it becomes $C \in \mathcal{C}_t$. Therefore, $C_{prev}$ describes exactly $C'$, the previous state of the same cluster. Now what remains is to show that $f_t$ is equivalent to $\mathcal{C}_t$. For any $C \in \mathcal{C}_t$, Algorithm 5 assigns the same id to all nodes of $C$, therefore we have $u \sim_C v$ if and only if $f_t(u) = f_t(v)$. ∎

**Lemma 9 (Lemma 8 in Cohen-Addad et al. (2022))** *Let $\epsilon$ be a small enough constant and $C, C'$ be non-singleton clusters of $\mathcal{C}_{t-1}^{agr}$ and $u, v$ two nodes in $C, C'$ respectively. In $\mathcal{C}_t^{agr}$, nodes $u, v$ cannot belong to the same cluster.*

## B.2. Deferred Proofs

We start by making a series of observations that help us understand under which conditions $r^{aux}(u)$ may increase. First, a node $u$ may be clustered in a non-singleton cluster of our solution $\mathcal{C}_t$ only if AGREEMENT "recommends" so, i.e., $u \in C \in \mathcal{C}_t^{agr} : |C| > 1$. Secondly, even if the latter is true our algorithm may decide to cluster $u$ as a singleton depending on the result of procedures MAKECONSISTENT and MAKEFAIR. Following our main algorithms notation let $\widetilde{C}, C^{fair}$ be the output of MAKECONSISTENT and MAKEFAIR respectively, and let $g_u^t$ to be the value of $g_u$ at the end of iteration $t$.

Note that:

1. $C^{fair}$ is either equal to $\widetilde{C}$ or empty.

2. $g_u^t = 0$ implies that $u \notin \widetilde{C}$. Thus, independently of whether $C^{fair}$ is equal to $\widetilde{C}$ or not, $u$ is clustered as a singleton at time $t$.

3. $g_u^t = 1$ implies that $u \in \widetilde{C}$. In that case, $u$ is clustered as a singleton if and only if $C^{fair} = \emptyset$.

We can deduce that, if $r^{aux}(u)$ increases at time $t$ then either: (a) $\{g_u^{t-1} = 1\} \wedge \{g_u^t = 0\}$; or (b) $\{g_u^{t-1} = 1\} \wedge \{g_u^t = 1\}$ and $C^{fair} = \emptyset$. At a high level, Lemma 2 and Lemma 3 provide bounds on the number of times $t$ when conditions (a) and (b) occur, respectively.

We now repeat the definition of an important event for completeness.

**Definition 1** *Let $C$ be a non-singleton cluster of $\mathcal{C}_t^{agr}$ and node $u_{C,t}$ be as calculated by Algorithm 4 at time $t$. If $|N_{G_t}(u_{C,t})| > (1 + 20\epsilon)|N_{G_{\tilde{t}_{u_{C,t}}}}(u_{C,t})|$ then we say that cluster $C$ and all its nodes participate in an important event at that time.*

Note that $\{g_u^{t-1} = 0\} \wedge \{g_u^{t-1} = 1\}$ implies that $u$ participates in an "important event" at time $t$. In addition, note that $|\{t : \{g_u^{t-1} = 1\} \wedge \{g_u^t = 0\}\}| \le |\{t : \{g_u^{t-1} = 0\} \wedge \{g_u^t = 1\}\}| + 1$. Thus, we can concentrate on upper bounding the number of important events that a node participates in.

**Lemma 2** *Let $t_1 < t_2$ be two times that $u$ participates in an important event. Then $|N_{G_{t_2}}(u)| \ge (1 + \epsilon)|N_{G_{t_1}}(u)|$*

**Proof** Let $C_1, C_2$ be the non-singleton clusters found by AGREEMENT that $u$ belongs to at times $t_1$ and $t_2$ respectively. W.l.o.g. we assume that $u$ does not participate in any other important event between times $t_1$ and $t_2$. In the following, for any node $v \in C$ we denote by $T(v, t)$ the value of variable $t_v$ at the beginning of iteration $t$. To ease notation we denote by $u^\star$ the node that provokes the *important event* for cluster $C_2$ at time $t_2$, i.e., $u^\star = u_{C_2, t_2}$. Also, we assume w.l.o.g. that $u_{C_2, t_2} \ne u$, otherwise we have that $|N_{G_{t_2}}(u)| > (1 + 20\epsilon)|N_{G_{T(u,t_2)}}(u)| \ge (1 + 20\epsilon)|N_{G_{t_1}}(u)|$ where the first inequality is true since $u = u^\star$ provokes the "important event" and the second inequality is true since $u$ participates at an "important event at time $t_1$, and consequently for any $t' \ge t_1$ we have $T(u, t') \ge t_1$, and in particular, $T(u, t_2) \ge t_1$. In addition, let $t_{in}^\star = T(t_2, u^\star)$ and denote by $C_{in}^\star \ni u^\star$ the cluster found by AGREEMENT at time $t_{in}^\star + 1$. Note that $C_{in}^\star$ is a non-singleton cluster. Indeed, if $u^\star$ was ever clustered as a singleton by AGREEMENT, the last time it may have happened is $t_{in}^\star$.

We now turn our attention to upper bounding $|N_{G_{t_2}}(u^\star) \setminus N_{G_{t_{in}^\star + 1}}(u^\star)|$. We proceed to show that at time $t_2 - 1$, $u^\star \in C_{t_2-1}^\star$ does not participate in an important event, where $C_{t_2-1}^\star \in \mathcal{C}_{t_2-1}^{agr}$.

Towards a contradiction we assume that $u^\star$ participates in an important event at iteration $t_2 - 1$ as well. Then at the end of that iteration we have $\widetilde{t}_u = t_2 - 1$ for all $u \in C^\star_{t_2-1}$. Since there is an important event at $t_2$ we have that:

$$|N_{G_{t_2-1}}(u_{C^\star_{t_2}}, t_2)| + 1 \geq |N_{G_{t_2}}(u_{C^\star_{t_2}}, t_2)| \geq (1 + 20\epsilon)|N_{G_{t_2-1}}(u_{C^\star_{t_2}}, t_2)|$$

where the first inequality is due to the degree of any node changing by at most 1 between two iterations and the second due to the important event at time $t_2$. But solving the inequality gives $|N_{G_{t_2-1}}(u_{C^\star_{t_2}}, t_2)| \leq \frac{1}{20\epsilon}$ which is a contradiction to the assumption that all degrees are above $\frac{1}{2\epsilon}$. Let $C_{t_2-1}$ be the non-singleton cluster to which $u^\star$ belongs according to the AGREEMENT at time $t_2 - 1$ and $u' = u_{C_{t_2-1}, t_2-1}$. Since $u^\star$ did not participate in an important event at time $t_2 - 1$ we have

$$|N_{G_{t_2-1}}(u')| < (1 + 20\epsilon)D(t_2 - 1, u').$$

At the same time since both $u'$ and $u^\star$ belong to the same non-singleton cluster in the agreement decomposition of time $t_2 - 1$ we can use [Property 7](#) and get

$$(1 - 5\epsilon)|N_{G_{t_2-1}}(u^\star)| \leq |N_{G_{t_2-1}}(u')|.$$

Also by the definition of $u'$, we have that

$$D(t_2 - 1, u') \leq D(t_2 - 1, u^\star) = |N_{G_{t^\star_{in}}}(u^\star)| \leq |N_{G_{t^\star_{in}+1}}(u^\star)|$$

Combining the last three inequalities we get:

$$|N_{G_{t_2-1}}(u^\star)| < \frac{(1 + 20\epsilon)|N_{G_{t^\star_{in}+1}}(u^\star)|}{(1 - 5\epsilon)} \overset{\text{for small enough } \epsilon}{\leq} (1 + 26\epsilon)|N_{G_{t^\star_{in}+1}}(u^\star)|.$$

Thus,

$$|N_{G_{t_2}}(u^\star) \setminus N_{G_{t^\star_{in}+1}}(u^\star)| \leq |N_{G_{t_2-1}}(u^\star) \setminus N_{G_{t^\star_{in}+1}}(u^\star)| + 1 \leq 26\epsilon|N_{G_{t^\star_{in}+1}}(u^\star)| + 1. \quad (\star\star)$$

We now consider two cases based on whether the intersection of $C^\star_{in}$ and $C_1$ is empty or non-empty.

1. If $C^\star_{in} \cap C_1 = \emptyset$ then we argue that most of the common neighbors of $u^\star$ and $u$ at time $t_2$ were not present at time $t_1$ and consequently $u$'s degree increased by more than a $(1 + \epsilon)$ multiplicative term. To argue the latter, it is enough to show that:

$$|N_{G_{t^\star_{in}+1}}(u) \setminus N_{G_{t_1}}(u)| + |N_{G_{t_2}}(u) \setminus N_{G_{t^\star_{in}}}(u)| \geq 2\epsilon|N_{G_{t_1}}(u)|. \quad (\star)$$

   Indeed, Eq. $(\star)$ implies that one of the two terms of the left-hand side is greater than $\epsilon|N_{G_{t_1}}(u)|$. Assume it is the first one (and the argument is the same if the second inequality is not satisfied), then $|N_{G_{t_2}}(u)| \geq |N_{G_{t^\star_{in}+1}}(u)| \geq |N_{G_{t_1}}(u)| + |N_{G_{t^\star_{in}+1}}(u) \setminus N_{G_{t_1}}(u)| \geq |N_{G_{t_1}}(u)| + \epsilon|N_{G_{t_1}}(u)| = (1 + \epsilon)|N_{G_{t_1}}(u)|$. Thus, we focus on proving Eq. $(\star)$ and to ease notation we denote the sum in the left-hand side as $I$.

   Since $u^\star \in C^\star_{in}$ at time $t^\star_{in} + 1$ and $u \in C_1$ at time $t_1$ we can apply [Property 10](#) and get:

$$|N_{G_{t_1}}(u) \setminus C_1| \leq 3\epsilon|N_{G_{t_1}}(u)|$$

$$|N_{G_{t^\star_{in}+1}}(u^\star) \setminus C^\star_{in}| \leq 3\epsilon|N_{G_{t^\star_{in}+1}}(u^\star)| \quad (\star\star\star)$$

combining the latter with $|N_{G_{t_{in}^\star+1}}(u) \setminus N_{G_{t_1}}(u)| \leq \epsilon|N_{G_{t_1}}(u)|$ helps us upper bound the term $|N_{G_{t_{in}^\star+1}}(u) \cap N_{G_{t_{in}^\star+1}}(u^\star)|$ as follows:

$$|N_{G_{t_{in}^\star+1}}(u) \cap N_{G_{t_{in}^\star+1}}(u^\star)| \leq |N_{G_{t_1}}(u) \cap N_{G_{t_{in}^\star+1}}(u^\star)| + |N_{G_{t_{in}^\star+1}}(u) \setminus N_{G_{t_1}}(u)|$$

$$\leq |C_1 \cap C_{in}^\star| + |N_{G_{t_1}}(u) \setminus C_1| + |N_{G_{t_{in}^\star+1}}(u^\star) \setminus C_{in}^\star| + |N_{G_{t_{in}^\star+1}}(u) \setminus N_{G_{t_1}}(u)|$$

$$\leq 0 + 3\epsilon|N_{G_{t_1}}(u)| + 3\epsilon|N_{G_{t_{in}^\star+1}}(u^\star)| + |N_{G_{t_{in}^\star+1}}(u) \setminus N_{G_{t_1}}(u)| \Rightarrow$$

$$|N_{G_{t_{in}^\star+1}}(u) \cap N_{G_{t_{in}^\star+1}}(u^\star)| \leq 3\epsilon(|N_{G_{t_1}}(u)| + |N_{G_{t_{in}^\star+1}}(u^\star)|) + |N_{G_{t_{in}^\star+1}}(u) \setminus N_{G_{t_1}}(u)|$$

$$(\star\star\star\star)$$

where in the third inequality we used that $C_1 \cap C_{in}^\star = \emptyset$ since we are in the first case and Eq. $(\star\star\star)$. Similarly we upper bound $|N_{G_{t_2}}(u) \cap N_{G_{t_2}}(u^\star)|$ as follows:

$$|N_{G_{t_2}}(u) \cap N_{G_{t_2}}(u^\star)|$$

$$\leq |N_{G_{t_{in}^\star+1}}(u) \cap N_{G_{t_{in}^\star+1}}(u^\star)| + |N_{G_{t_2}}(u) \setminus N_{G_{t_{in}^\star+1}}(u)| + |N_{G_{t_2}}(u^\star) \setminus N_{G_{t_{in}^\star+1}}(u^\star)|$$

$$\leq 3\epsilon(|N_{G_{t_1}}(u)| + |N_{G_{t_{in}^\star+1}}(u^\star)|) + I + 26\epsilon|N_{G_{t_{in}^\star+1}}(u^\star)| + 1$$

$$\leq 32\epsilon \max\{|N_{G_{t_2}}(u)|, |N_{G_{t_2}}(u^\star)|\} + 1 + I \xRightarrow{Property7}$$

$$(1 - 5\epsilon)\max\{|N_{G_{t_2}}(u)|, |N_{G_{t_2}}(u^\star)|\} \leq 32\epsilon\max\{|N_{G_{t_2}}(u)|, |N_{G_{t_2}}(u^\star)|\} + 1 + I \Rightarrow$$

$$I \geq (1 - 37\epsilon)\max\{|N_{G_{t_2}}(u)|, |N_{G_{t_2}}(u^\star)|\} - 1 \Rightarrow$$

$$I \geq (1 - 37\epsilon)|N_{G_{t_1}}(u^\star)| - 1 \xRightarrow{\epsilon \text{ small enough}}$$

$$I \geq 2\epsilon|N_{G_{t_1}}(u^\star)|$$

where in the second inequality we used Eq. $(\star\star\star\star)$ and Eq. $(\star\star)$.

2. If $C_{in}^\star \cap C_1 \neq \emptyset$ then let $v \in C_{in}^\star \cap C_1$ be a node which at time $t_1$ is in the same cluster as $u$ and at time $t_{in}^\star + 1$ is in the same cluster as $u^\star$. The following series of inequalities provides a rough sketch of the formal proof. We will use the definition of an "important" event and the observation that nodes within the same cluster of the agreement decomposition have similar neighborhoods:

$$|N_{G_{t_2}}(u)| \overset{u,u^\star \in C_2}{\simeq} |N_{G_{t_2}}(u^\star)| \overset{\text{"important" event}}{>} (1 + 20\epsilon)|N_{G_{t_{in}^\star}}(u^\star)| \simeq (1 + 20\epsilon)|N_{G_{t_{in}^\star+1}}(u^\star)|$$

$$\overset{u^\star,v \in C_{in}^\star}{\simeq} (1 + 20\epsilon)|N_{G_{t_{in}^\star+1}}(v)| \geq (1 + 20\epsilon)|N_{G_{t_1}}(v)| \overset{u,v \in C_1}{\simeq} (1 + 20\epsilon)|N_{G_{t_1}}(u)|.$$

We now proceed to formally prove the latter.

From Property 7 we have that:

$$|N_{G_{t_1}}(u) \cap N_{G_{t_1}}(v)| \geq (1 - 5\epsilon)\max\{|N_{G_{t_1}}(u)|, |N_{G_{t_1}}(v)|\}$$

$$|N_{G_{t_{in}^\star+1}}(u^\star) \cap N_{G_{t_{in}^\star+1}}(v)| \geq (1 - 5\epsilon)\max\{|N_{G_{t_{in}^\star+1}}(u^\star)|, |N_{G_{t_{in}^\star+1}}(v)|\}.$$

From this we get that:

$$|N_{G_{t_1}}(v)| \geq (1 - 5\epsilon)|N_{G_{t_1}}(u)|$$
$$|N_{G_{t_{in}^\star+1}}(u^\star)| \geq (1 - 5\epsilon)|N_{G_{t_{in}^\star+1}}(v)|.$$

Combining the last two inequalities with the facts that $|N_{G_{t_{in}^\star+1}}(v)| \geq |N_{G_{t_1}}(v)|$ as well as $|N_{G_{t_{in}^\star}}(u^\star)| + 1 \geq |N_{G_{t_{in}^\star+1}}(u^\star)|$ we conclude that:

$$|N_{G_{t_1}}(v)| \geq (1 - 5\epsilon)|N_{G_{t_1}}(u)|$$
$$|N_{G_{t_{in}^\star}}(u^\star)| \geq (1 - 5\epsilon)^2|N_{G_{t_1}}(u)| - 1. \qquad (*)$$

At time $t_2$ nodes $u$ and $u^\star$ are in the same cluster $C_2$. Thus, using [Property 7](#) we have that:

$$|N_{G_{t_2}}(u^\star) \cap N_{G_{t_2}}(u)| \geq (1 - 5\epsilon) \max\{|N_{G_{t_2}}(u^\star)|, |N_{G_{t_1}}(v)|\} \Rightarrow$$
$$|N_{G_{t_2}}(u)| \geq (1 - 5\epsilon)|N_{G_{t_2}}(u^\star)| \Rightarrow$$
$$|N_{G_{t_2}}(u)| \geq (1 - 5\epsilon)(1 + 20\epsilon)|N_{G_{t_{in}^\star}}(u^\star)| \Rightarrow$$
$$|N_{G_{t_2}}(u)| \geq (1 - 5\epsilon)(1 + 20\epsilon)\left((1 - 5\epsilon)^2|N_{G_{t_1}}(u)| - 1\right)$$
$$|N_{G_{t_2}}(u)| \geq (1 + \epsilon)|N_{G_{t_1}}(u)|$$

where the last inequality holds for $\epsilon$ small enough.

∎

## Appendix C. Bound the Competitive Ratio

The main idea behind the proof of Lemma [5](#) is to use a charging argument in Lemma [10](#) and Lemma [11](#) to show that the cost MAKECONSISTENT incurs by splitting clusters of AGREEMENT is within a constant of the optimal cost.

Before we proceed with the statements, we introduce some notation that helps indicate which nodes have been singletons in the past and when. For any $C \in \mathcal{C}_t^{agr}$ we define indicator $s(u, t)$ that is $1$ only if $u$ has been clustered as a singleton by AGREEMENT in some iteration before $t$, formally $s(u, t) = \mathbb{I}\{\exists t' < t : u \in \mathcal{S}(\mathcal{C}_{t'}^{agr})\}$. We also define as $T(t, u), D(t, u)$ as the last such time and degree of $u$ at that time respectively:

$$T(t, u) = \begin{cases} \max\{t' \leq t : u \in \mathcal{S}(\mathcal{C}_{t'}^{agr})\} & \text{if } s(u, t) = 1 \\ \infty & \text{otherwise} \end{cases}$$
$$D(t, u) = |N_{G_{T(t,u)}}(u)|.$$

Now we define the nodes of a cluster that have label $0$ at the end of an iteration as $Z_C = \{u \in C : g(u) = 0\}$ and further define $Z_C^s = \{u \in Z_C : T(t, u) = \tilde{t}_u\}$, i.e. the set of nodes $u$ with label $g_u = 0$ which have been singleton in the agreement decomposition of time $T(u, t)$ and have not participated in an important event since then. These are exactly the nodes that will allow us to

charge the algorithm's cost to the optimal solution. We further define as $\mathcal{Z}_t^s$ all the nodes $u$ that have label $g_u = 0$ and have been singletons in the past:

$$\mathcal{Z}_t^s = \bigcup_{C \in \mathcal{C}_t^{agr}} Z_C^s.$$

Now we are ready to state the two charging lemmas, which resemble Lemma 3.7 and 3.8 of Cohen-Addad et al. (2021). The crucial difference is that we charge edges at a previous time and collect the debt at the current time $t$ to charge it to either $\mathcal{O}_t$ or $\mathcal{C}_t^{agr}$.

**Lemma 10** *The edges adjacent to all vertices $u \in \mathcal{Z}_t^s$ which (1) are not paid for by $\mathcal{O}_t$, (2) have endpoints in agreement at time $t$ and (3) were removed at time $T(t, u)$ by Line 1 of Algorithm 1 are at most $\frac{4}{\epsilon(1-\epsilon)} \operatorname{cost}(\mathcal{O}_t)$.*

**Lemma 11** *The edges adjacent to all vertices $u \in \mathcal{Z}_t^s$ which (1) are not paid for by $\mathcal{O}_t$, (2) have endpoints in agreement at time $t$ and (3) were removed at time $T(t, u)$ by Line 2 of Algorithm 1 are at most $\frac{2}{\epsilon(1-\epsilon)^3} \operatorname{cost}(\mathcal{C}_t^{agr}) + (\frac{4}{\epsilon(1-\epsilon)^3} + \frac{2}{\epsilon^2(1-\epsilon)^3}) \operatorname{cost}(\mathcal{O}_t)$.*

Intuitively the charging arguments work because (1) in Lemma 12 we show that there are sufficiently many nodes in each cluster that have been singleton in some "recent" iteration; and (2) as shown in Lemma 13 the degrees of these nodes have not changed significantly since the last time AGREEMENT clustered them as singletons.

The next lemma formalizes (1), by showing that whenever MAKECONSISTENT splits a cluster $C \in \mathcal{C}_t^{agr}$ into singletons $Z_C$ and a cluster $C \setminus Z_C$ there are enough nodes in $C$ which have been singleton in the "recent" past, as these are nodes to which we can charge the cost incurred.

**Lemma 12** *Let $C \in \mathcal{C}_t^{agr}$ then $|Z_C^s| > \frac{\epsilon}{100}|Z_C|$.*

We defer the proof to Appendix D as it is highly technical. We now proceed to show (2), i.e., that the degree of any node $u \in Z_C^s$ has not changed since time $\widetilde{t}_u$.

**Lemma 13** *Let $\epsilon$ be an adequately small constant. Let $C \in \mathcal{C}_t^{agr}$ that does not participate in an important event at time $t$, then for any $u \in Z_C^s$ we have $|N_{G_t}(u)| \leq 2 \cdot D(t, u)$.*

Note that at each time $t$, FAIR-CONSISTENT-AGREEMENT may only modify non-singleton clusters of $\mathcal{C}_t^{agr}$. Consequently, only those modifications may increase the cost incurred by FAIR-CONSISTENT-AGREEMENT with respect to AGREEMENT, and the goal of this section is to bound such increase. Such modifications are achieved by further splitting each non-singleton cluster $C \in \mathcal{C}_t^{agr}$ using the subroutines MAKECONSISTENT and MAKEFAIR sequentially. Thus, in the analysis, we first prove that only applying MAKECONSISTENT to each non-singleton cluster of $\mathcal{C}_t^{agr}$ is constant competitive to the optimal unconstrained clustering $\mathcal{O}_t$ on every step $t$. Before we state the lemma we need some auxiliary notation that captures the clustering produced by MAKECONSISTENT across all clusters of AGREEMENT. Let $\widetilde{C}(t, C)$ be the output of running MAKECONSISTENT on cluster $C \in \mathcal{C}_t^{agr}$ as in Algorithm 2 and $Z_C = C \setminus \widetilde{C}(t, C)$. We denote by $\widetilde{\mathcal{C}}_t$ the clustering produced by running MAKECONSISTENT across all $C \in \mathcal{C}_t^{agr}$, which is formally defined as

$$\widetilde{\mathcal{C}}_t = \{\widetilde{C}(t, C) : C \in \mathcal{C}_t^{agr}\} \bigcup \{\{u\} : u \in \bigcup_{C \in \mathcal{C}_t^{agr}} Z_C\}.$$

**Lemma 5** *For a constant $\epsilon$ small enough we have* $\mathrm{cost}(\widetilde{\mathcal{C}}_t) \leq \Theta\left(\frac{1}{\epsilon^3}\right) \cdot \mathrm{cost}(\mathcal{O}_t)$ *for any $t$.*

**Proof** Let $C \in \mathcal{C}_t^{agr}$. Unless it participates in an important event at time $t$, MAKECONSISTENT splits $C$ into singletons $Z_C$ and a new cluster $C \setminus Z_C$. Thus, for every node $u \in Z_C$ we pay at most 1 for every one of its adjacent edges at time $t$. Then we have:

$$\mathrm{cost}(\widetilde{\mathcal{C}}_t) \leq \mathrm{cost}(\mathcal{C}_t^{agr}) + \sum_{C \in \mathcal{C}_t^{agr}} \sum_{u \in Z_C} |N_{G_t}(u)| \tag{0}$$

We now show that the cost incurred by splitting the nodes in $Z_C$ is only a constant away from that of splitting $Z_C^s$, i.e. $\sum_{u \in Z_C} |N_{G_u}(t)| \leq O(1) \sum_{u \in Z_C^s} |N_{G_t}(u)|$. To do that we charge the cost incurred by each node in $Z_C \setminus Z_C^s$ to that of nodes in $Z_C^s$. Intuitively this argument works because the degrees of nodes in a cluster are nearly equal and $Z_C^s$ is large enough. Let $u \in Z_C \setminus Z_C^s$, then $u$ charges $\frac{|N_{G_t}(u)|}{|Z_C^s|}$ to each $v \in Z_C^s$. Then in total $u$ has charged

$$\frac{|N_{G_t}(u)|}{|Z_C^s|} \cdot |Z_C^s| = |N_{G_t}(u)| \geq (1 - 9\epsilon)|C| \tag{1}$$

where the inequality is due to Property 6. Now let $v \in Z_C^s$, then it has been charged by every $u \in Z_C$ for a total of

$$\begin{aligned}
\frac{\sum_{u \in Z_C \setminus Z_C^s} |N_{G_t}(u)|}{|Z_C^s|} &\leq \frac{\sum_{u \in Z_C \setminus Z_C^s} \frac{1}{1-3\epsilon} \cdot |C|}{|Z_C^s|} \\
&\leq \frac{|Z_C| \frac{1}{1-3\epsilon} \cdot |C|}{|Z_C^s|} \\
&\leq \frac{|Z_C| \frac{1}{1-3\epsilon} \cdot |C|}{\frac{\epsilon}{100}|Z_C|} \\
&= \frac{100}{\epsilon(1-3\epsilon)}|C| \tag{2}
\end{aligned}$$

where the first inequality is due to Property 10, the second inequality is due to $Z_C \setminus Z_C^s \subseteq Z_C$ and the third inequality is due to Lemma 12. Combining Equations (1) and (2) and summing we have that

$$\sum_{u \in Z_C} |N_{G_t}(u)| \leq \sum_{u \in Z_C \setminus Z_C^s} |N_{G_t}(u)| \leq \frac{100}{\epsilon(1-3\epsilon)(1-9\epsilon)} \sum_{u \in Z_C^s} |N_{G_t}(u)| \leq \frac{400}{\epsilon} \sum_{u \in Z_C^s} |N_{G_t}(u)| \tag{3}$$

where the last inequality is due to $1 - 9\epsilon \leq \frac{1}{2}$ for $\epsilon$ small enough. We proceed with bounding the cost from Equation (0):

$$\begin{aligned}
\mathrm{cost}(\widetilde{\mathcal{C}}_t) &\leq \mathrm{cost}(\mathcal{C}_t^{agr}) + \sum_{C \in \mathcal{C}_t^{agr}} \frac{400}{\epsilon} \cdot \sum_{u \in Z_C^s} |N_{G_t}(u)| \\
&\leq \mathrm{cost}(\mathcal{C}_t^{agr}) + \sum_{C \in \mathcal{C}_t^{agr}} \frac{400}{\epsilon} \cdot \sum_{u \in Z_C^s} 2 \cdot D(t,u) \\
&= \mathrm{cost}(\mathcal{C}_t^{agr}) + \frac{400}{\epsilon} \cdot \sum_{C \in \mathcal{C}_t^{agr}} \sum_{u \in Z_C^s} D(t,u)
\end{aligned}$$

where the first inequality is due to Equation (3) and the second due to Lemma 13. Since by Lemma 3 it holds that $\mathrm{cost}(\mathcal{C}_t^{agr}) \leq \Theta(1)\,\mathrm{cost}(\mathcal{O}_t)$, to prove the current lemma it is sufficient to argue that $\sum_{u \in Z_t^s} D(t,u) \leq \Theta(1)\,\mathrm{cost}(\mathcal{O}_t)$. Note that for any node $u \in Z_t^s$, at time $T(t,u)$ it was singleton in $\mathcal{C}_{T(t,u)}^{agr}$ and had $D(t,u)$ neighbors by definition. For it to be singleton all its edges $(u,v)$ must have been removed by Algorithm 1 so we can proceed as follows:

1. All edges $(u,v)$ that are paid for by $\mathcal{O}_t$ can be charged to it for a total of $\mathrm{cost}(\mathcal{O}_t)$.

2. All edges $(u,v)$ that are paid for by $\mathcal{C}_t^{agr}$ can be charged to it for a total of $\mathrm{cost}(\mathcal{C}_t^{agr})$.

3. All edges $(u,v)$ that do not belong to categories (1) or (2) and were deleted by Line 1 of Algorithm 1 at $T(t,u)$ can be charged to $\mathcal{O}_t$ with Lemma 10 for a total of $\frac{4}{\epsilon(1-\epsilon)}\mathrm{cost}(\mathcal{O}_t)$.

4. All edges $(u,v)$ that do not belong to categories (1), (2) or (3) were deleted by Line 2 of Algorithm 1 at $T(t,u)$ and can be charged to $\mathcal{C}_t^{agr}, \mathcal{O}_t$ with Lemma 11 for a total of $\frac{2}{\epsilon(1-\epsilon)^3}\mathrm{cost}(\mathcal{C}_t^{agr}) + (\frac{4}{\epsilon(1-\epsilon)^3} + \frac{2}{\epsilon^2(1-\epsilon)^3})\mathrm{cost}(\mathcal{O}_t)$.

∎

**Proof of Lemma 10** Take a $u \in \mathcal{Z}_t^s$ at time $t$ and one of its adjacent edges $(u,v)$ that was present at $T(t,u)$. To ease notation we define $d_t(u) = |N_{G_t}(u)|$. Notice that $u$ was singleton in $\mathcal{C}_{T(u,t)}^{agr}$ as such a time exists that it was singleton by definition of $Z_t^s$. We can restate the properties given by the assumption as follows:

1. $\mathcal{O}_t$ does not pay for $(u,v)$ therefore it clusters $u,v$ together.

2. At time $t$ $u,v$ are in $\epsilon$-agreement which implies that their degrees are very close:
$$\max\{d_t(u), d_t(v)\} \leq \frac{1}{1-\epsilon}d_t(u) \leq \frac{2}{1-\epsilon}D(t,u)$$
where the second inequality is due to Lemma 13.

3. At time $T(t,u)$ they must have not been in $\epsilon$-agreement since Line 1 deleted their edge. So $|N_{G_{T(t,u)}}(u) \triangle N_{G_{T(t,u)}}(v)| > \epsilon\max\{d_{T(t,u)}(u), d_{T(t,u)}(v)\}$ and this implies there are many nodes $w$ which have an edge with only $u$ or only $v$. This property remains true at time $t$ since nothing has changed in the triangle $u,v,w$. Then $\mathcal{O}_t$ must pay for either $(u,w)$ or $(v,w)$ since it has clustered $u$ and $v$ together.

Using these properties we can formulate the charging argument as follows. Every such edge $(u,v)$ charges $\frac{1}{\epsilon\max\{d_t(u), d_t(v)\}}$ to each of the neighboring edges/non-edges $(u,w)$ or $(v,w)$ that $\mathcal{O}_t$ pays for. Then each $(u,v)$ has charged for a total of
$$\frac{\epsilon\max\{d_{T(t,u)}(u), d_{T(t,u)}(v)\}}{\epsilon\max\{d_t(u), d_t(v)\}} \geq \frac{D(t,u)}{\frac{2}{1-\epsilon}D(t,u)} = \frac{1-\epsilon}{2}.$$

If we consider an edge/non-edge $(a,b)$ that $\mathcal{O}_t$ pays for then only neighboring edges charge it. Each of the neighboring edges to $a$ charge it at most by $\frac{1}{\epsilon d_t(a)}$ and similarly for $b$ so the total charge is at most
$$\frac{1}{\epsilon d_t(a)} \cdot d_t(a) + \frac{1}{\epsilon d_t(b)} \cdot d_t(b) = \frac{2}{\epsilon}.$$

Overall, since by definition of the charging argument, only edges paid for by $\mathcal{O}_t$ are charged, we have paid $\frac{4}{\epsilon(1-\epsilon)}$ for every edge that $\mathcal{O}_t$ paid. ∎

**Proof of Lemma 11** Take a $u \in \mathcal{Z}_t^s$ at time $t$ and one of its adjacent edges $(u, v)$ that was present at $T(t, u)$. To ease notation we define $d_t(u) = |N_{G_t}(u)|$. Notice that $u$ was singleton in $\mathcal{C}_{T(u,t)}^{agr}$ by definition of $\mathcal{Z}_t^s$. As before we know by the assumption that $\mathcal{O}_t$ clusters $u, v$ together and that they are in $\epsilon$-agreement at time $t$ so their degrees can be at most $\frac{1}{1-\epsilon}$ away by definition of agreement. But now we also know that at time $T(t, u)$ they were in agreement and they were both light so for $y \in \{u, v\}$ we have that $y$ disagrees with $\epsilon d_{T(t,u)}(y)$ of its neighbors which we denote $v_1, \ldots, v_{\epsilon d_{T(t,u)}(y)}$. We can proceed with charging:

- If $(y, v_i)$ is not paid for by $\mathcal{O}_t$ we distinguish between the following cases:
    - If $y, v_i$ disagree at time $t$ then $(u, v)$ charges $\frac{1}{2\epsilon d_t(y)}$ to edge $(y, v_i)$ that will be paid by $\mathcal{C}_t^{agr}$ since it cuts this edge and is denoted **purple debt**.
    - If $y, v_i$ agree at time $t$, then we use that at time $T(t, u)$, $y$ and $v_i$ disagree. Thus there exist $\epsilon \max\{d_{T(t,u)}(y), d_{T(t,u)}(v_i)\}$ triangles $(y, v_i, w)$ so that $\mathcal{O}_t$ pays for either $(y, w)$ or $(v_i, w)$:
        * If $\mathcal{O}_t$ pays for edge/non-edge $(v_i, w)$ then $(u, v)$ charges $\frac{1}{2\epsilon^2 d_t(y) d_t(v_i)}$ to it. We call such charge, **blue debt** and note that $\mathcal{O}_t$ will pay for that.
        * If $\mathcal{O}_t$ pays for edge/non-edge $(y, w)$ then $(u, v)$ charges $\frac{1}{2\epsilon^2 d_t(y)^2}$ to it .We call such charge, **red debt** and note that $\mathcal{O}_t$ will pay for that.

- If $(y, v_i)$ is paid for by $\mathcal{O}_t$ then edge $(u, v)$ charges $\frac{1}{2\epsilon d_t(y)}$ to it that will be paid by $\mathcal{O}_t$ and is denoted **green debt**.

By definition only edges/non-edges paid by $\mathcal{O}_t$ or $\mathcal{C}_t^{agr}$ are charged. The proof proceeds by lower bounding the amount of charge that an edge $(u, v)$ distributes and upper bounding the amount of charge that any edge/non-edge paid by either $\mathcal{O}_t$ or $\mathcal{C}_t^{agr}$ receives. We start with the former lower bound.

In our charging scheme, notice that for $y \in \{u, v\}$ if $(y, v_i)$ is not cut by $\mathcal{O}_t$ either blue or red debt is at least $\frac{1}{2\epsilon^2 d_t(y) \max\{d_t(y), d_t(v_i)\}}$ per triangle so each edge $(y, v_i)$ causes $(u, v)$ to charge

$$\frac{\epsilon \max\{d_{T(t,u)}(y), d_{T(t,u)}(v_i)\}}{2\epsilon^2 d_t(y) \max\{d_t(y), d_t(v_i)\}} = \frac{1}{2\epsilon d_t(y)} \frac{\max\{d_{T(t,u)}(y), d_{T(t,u)}(v_i)\}}{\max\{d_t(y), d_t(v_i)\}}$$
$$\geq \frac{1}{2\epsilon d_t(y)} \frac{(1 - \epsilon^3)}{2}$$
$$= \frac{(1 - \epsilon)^3}{4\epsilon d_t(y)}$$

where the inequality is due to $u, v$ agreeing in $T(t, u)$ which makes the numerator greater than $(1 - \epsilon)D(t, u)$ and $u, v, v_i$ agreeing at $t$ in the case of blue debt so the divisor is at most $\frac{d_t(u)}{(1-\epsilon)^2} \leq \frac{2D(t,u)}{(1-\epsilon)^2}$ due to Lemma 13. Then green or purple debt charge at least that much per $(y, v_i)$ edge, and since there are two choices for $y$ and $\epsilon d_{T(t,u)}(y)$ such edges the total charge is at least $\frac{(1-\epsilon)^3}{2}$.

Now we can count the total debt charged to the edges cut by $\mathcal{O}_t$ and $\mathcal{C}_t^{agr}$. Consider edge/non-edge $(a, b)$ paid for by $\mathcal{O}_t$ and we have the following cases:

- **Blue debt**: Edge/non-edge $(a, b)$ is of the form $(v_i, w)$. For half the debt assume $v_i \equiv a$ then it can be charged at most for each neighbor of $y$ where $y$ is any of $a's$ neighbors so $\frac{1}{2\epsilon^2 d_t(y) d_t(v_i)} \cdot d_t(y) \cdot d_t(v_i)$. Overall that is at most $\frac{1}{\epsilon^2}$ debt.

- **Red debt**: Edge/non-edge $(a, b)$ is of the form $(y, w)$. For half the debt assume $y \equiv a$ then this edge is charged at most for each of its neighbors it disagrees with, for each possible debt charging edge $(u, v)$ so $\frac{1}{2\epsilon^2 d_t(a)^2} \cdot \epsilon d_t(a) \cdot d_t(a)$. Overall that is at most $\frac{1}{\epsilon}$ debt.

- **Green debt**: Edge $(a, b)$ is of the form $(y, v_i)$. For half the debt assume $y \equiv a$ then it is charged at most once for each of its neighbors so $\frac{1}{2\epsilon d_t(a)} \cdot d_t(a)$. Overall that is at most $\frac{1}{\epsilon}$ debt.

Consider edge $(a, b)$ cut by $\mathcal{C}_t^{agr}$:

- **Purple debt**: Edge $(a, b)$ is of the form $(y, v_i)$. For half the debt assume $y \equiv a$ then it is charged at most for each neighbor of $a$ so $\frac{1}{2\epsilon d_t(y)} \cdot d_t(y)$. Overall that is $\frac{1}{\epsilon}$ debt.

Overall for each edge/non-edge paid for by $\mathcal{O}_t$ we paid at most $\frac{2/\epsilon + 1/\epsilon^2}{(1-\epsilon)^3/2} = \frac{4}{\epsilon(1-\epsilon)^3} + \frac{2}{\epsilon^2(1-\epsilon)^3}$ and for each edge cut by $\mathcal{C}_t^{agr}$ we paid at most $\frac{1/\epsilon}{(1-\epsilon)^3/2} = \frac{2}{\epsilon(1-\epsilon)^3}$. ∎

**Proof** of Lemma 13 We know that $u$ has a similar degree with $u_{g,C,t}$ as they are both clustered in $C$ at time $t$ by AGREEMENT, but we also know that by definition $u_{C,t}$ had the lowest degree last time it was a singleton so we have:

$$|N_{G_t}(u)| \leq \frac{1}{1 - 5\epsilon} \cdot |N_{G_t}(u_{C,t})| \leq \frac{1 - 100\epsilon}{1 - 5\epsilon} \cdot D(t, u_{C,t}) \leq \frac{1 - 100\epsilon}{1 - 5\epsilon} \cdot D(t, u) \leq 2 \cdot D(t, u)$$

where the first inequality is due to Property 8, the second inequality due to not having an important event therefore the condition in line 6 of Algorithm 4 is false, the third inequality by definition of $u_{C,t}$ and the fourth due to $\epsilon$ being adequately small. ∎

# Appendix D. Dynamic Analysis of the Clustering Sequence $\mathcal{C}_1^{agr}, \mathcal{C}_2^{agr}, \ldots$

To bound the competitive ratio we need to first understand the clustering structure of the MAKE-CONSISTENT algorithm. This inevitably leads us to examine how a cluster may evolve in the recommendations given by the agreement algorithm.

**Definition 4 (history of a cluster)** *Given a cluster $C_{t_l} \in \mathcal{C}_{t_l}^{agr} v$ we say that the sequence of clusters $C_{t_1}, \ldots, C_{t_l}$ is the "history" of $C_{t_l}$ and than $t_1$ is the formation time of $C_{t_l}$*

*1. $C_{t_i} \in \mathcal{R}(\mathcal{C}_{t_i}^{agr})$, for all $i \in [l]$; and*

*2. $C_i \cap C_{i+1} \neq \emptyset$ for all $i \in [l-1]$; and*

*3. $C_{t_1}, \ldots, C_{t_l}$ is the sequence of maximum length for which (1) and (2) hold.*

**Lemma 14** *The history of any cluster is unique.*

**Proof** Let $C \in \mathcal{C}_t^{agr}$ and its history $C_{t_C}, C_{t_C+1}, \ldots, C_t = C$. We will argue inductively that for any $t' \in t_C, \ldots, t$ $C_{t'}$ is uniquely defined by the requirement that consecutive clusters have nonempty intersections. The base case $t' = t$ is obvious as $C_t = C$. Then for the inductive step, we argue that $C_{t'-1}$ is unique. By the definition of the history we have $C_{t'-1} \cap C_{t'} \neq \emptyset$. Notice that both clusters were the output of AGREEMENT at consecutive times, therefore by Lemma 9 we conclude the induction. ∎

**Definition 5 (Split event)** *We have a "split event" for node $C$ at time $t$ if at the end of Algorithm 4 we have $Z_{C_t} = C_t$ and the cluster is split into singletons.*

**Definition 6 (Split event indicators)** *When it is clear from context that we focus on the history of $C$: $C_{t_C}, \ldots, C_t$, we define $E_{t'} = \{Z_{C_{t'}} \leq 100\epsilon|C|\}$ where $Z_{C_{t'}}$ are the nodes $u : g(u) = 0$ at the end of iteration $t'$.*

**Lemma 15** *Let $E_{t'}$ be the indicator function that refers to the history of some cluster $C \in \mathcal{C}_t^{agr}$. Then $E_{t'}$ is false if and only if there is a split event.*

**Proof** We start by assuming that $E_{t'}$ is false. Then cluster $C_{t'}$ must have particiapated in a split event because line 6 of Algorithm 4 can only decrease the number of nodes $u$ with $g(u) = 0$, therefore in line 2 of Algorithm 4 the condition was true. Now suppose $E_{t'}$ is true. Cluster $C_{t'}$ cannot have participated in a split event either because the condition in line 2 of Algorithm 4 was false, or because it was true and an important event kept the cluster together. ∎

**Definition 7 (Transition event)** *We have a "transition event" at time $t$ if $E_{t-1} \wedge \overline{E_t}$ is true.*

**Lemma 12** *Let $C \in \mathcal{C}_t^{agr}$ then $|Z_C^s| > \frac{\epsilon}{100}|Z_C|$.*

**Proof** Assume towards a contradiction that $|Z_{C_t}^s| \leq \frac{\epsilon}{100}|Z_{C_t}|$. Denote by $t_C = t_{C_t}$ the cluster formation time of $C_t$ and let $C_{t_C}, C_{t_C+1}, \ldots, C_t$ be the "history" of $C_t$.

**Overview** We make a series of observations:

1. $\exists t' \in \{t_C, \ldots, t\} : \overline{E_{t'}}$ otherwise we would have that $Z_{C_t}^s = Z_{C_t}$.

   From now on denote $t^*$ as the maximum time in $\{t_C, \ldots, t\}$ such that $E_{t^*-1} \wedge \overline{E_{t^*}}$, i.e. a transition event. We proceed to study the events that occur after $t^*$.

2. No important events occur between $t^*$ and $t$.

3. Split events occur on every iteration between $t*$ and $t$.

   Now we turn to the iterations before $t^*$ and informally we show that between $t_C$ and $t^*$ the nodes that join the cluster without ever being singleton are not too many.

4. Between any consecutive transition events $t_1, t_2$ there is an important event $t_I$. Furthermore, between $t_I$ and $t_2$, there are no split events.

5. Using the previous observation we argue that $|\{u \in C_{t^*} : s(u, t^*) = 0\}| < (1 - 100\epsilon)|C_{t^*}|$.

Using these observations we can conclude the contradiction. The intuition is that due to the assumption at $t$, many more nodes have never been singleton. Since there were significantly fewer at time $t^*$ that implies many of them must have joined between $t^*$ and $t$. But once they join they cannot change cluster (they would have to become singleton first). Therefore the first of them to arrive must have increased its degree significantly between the time it arrived and $t$, as it remained clustered with them for the whole duration. But that would imply an important event. We now proceed to prove the observations in the order they were stated:

**Observation 1** Suppose that there were no $t' \in \{t_C, \ldots, t\} : \overline{E_{t'}}$. Then the condition on line 2 of Algorithm 4 would always be false and any node $u$ with $g(u) = 0$ for $t_C < t'' \leq t$ would have acquired this label due to line 8 of Algorithm 2 (at no other point does the algorithm assign $g(u) = 0$). We also know that $Z^s_{C_{t_C}} = Z_{C_{t_C}}$ since by definition of $t_C$, its nodes were singletons due to AGREEMENT for all previous times. This implies $Z^s_{C_t} = Z_{C_t}$ which is a contradiction.

**Observation 2** Assume that an important event does happen before $t$ and denote the first such time after $t^*$ as $t_I \in \{t^*, \ldots, t\}$. Then at the end of MAKECONSISTENT we have $Z_{C_{t_I}} = \emptyset$. But that would imply $|Z^s_{C_{t_I}}| = |Z_{C_{t_I}}| = 0$ and $E_{t_I}$ is true. Due to the maximality of $t^*$ we know that for any $t' \in \{t_I, \ldots, t\} : E_{t'}$. We will now show that this implies $Z^s_{C_t} = Z_{C_t}$ which is a contradiction to our initial assumption. In fact we can show inductively that $Z^s_{C_{t'}} = Z_{C_{t'}}$ for any $t' \in \{t_I, \ldots, t\}$. Note that any node $u \in Z_{C_{t_I}}$ before the important event had its label $g(u) = 1$ which gives the base case. For the inductive step consider time $t' + 1$ and denote $Z^{start}, Z^{end}$ as the set $Z_{C_{t'+1}}$ at the beginning and the end of the execution of Algorithm 4. The only nodes in $Z^{start} \setminus Z_{C_{t'}}$ are those that $\mathcal{C}^{agr}_{t'+1}$ brought into the cluster. These must have been singletons at time $t'$ since AGREEMENT cannot join different clusters. Therefore they only contribute to $Z^s_{C_{t'}}$. Furthermore, notice that $Z^{end} \subseteq Z^{start}$ as either the condition in line 2 of Algorithm 4 was false, or it was true and the condition at line 5 was true as well, which gave an important event and $Z^{end} = \emptyset$. These are the only possible cases as $E_{t'+1}$ is true. Therefore $Z^s_{C_{t'+1}} = Z_{C_{t'+1}}$ and that concludes the inductive step and the contradiction.

**Observation 3** Due to maximality of $t^*$ there are only two cases for the values of $E_{t'}$ for $t' \in \{t^*, \ldots, t\}$. Either they are all false, or there is a switching time $t_s$ such that $\bigwedge^{t_s}_{t'=t^*} \overline{E_{t'}}$ is true and $\bigwedge^t_{t'=t_s+1} E_{t'}$ is true. Notice that in the second case, there must be an important event at time $t_s + 1$ which would contradict observation 3. We assume that $t_s + 1$ is not an important event towards a contradiction. Since $C_{t_s}, C_{t_s+1}$ are consecutive clusters in the history of $C$ by Lemma 18 we have:

$$|C_{t_s} \setminus C_{t_s+1}| \leq 18\epsilon|C_{t_s}|. \tag{*}$$

We define $Z$ as $Z_{C_{t_S}}$ at the end of Algorithm 4 and notice that $Z = C_{t_s}$ since $\overline{E_{t_s}}$ is true. We also define $Z^{start}, Z^{end}$ as $Z_{C_{t_s+1}}$ at the start and end of Algorithm 4 and notice that since $E_{t_s+1}$ is true and $t_s + 1$ is not an important event we have $|Z^{start}| \leq |Z^{end}| \leq 100\epsilon|C_{t_s+1}|$. Notice that if $u \in Z \setminus Z^{start}$ then AGREEMENT must have removed it from the cluster and set it as a singleton since there was no important event to change its label $g(u)$ from 0 to 1. Furthermore, for any node $u \in C_{t_s+1} : g(u) = 1$ we must have that $u \in C_{t_s}$ (except for the node that arrived at $t_s + 1$), as no nodes from other clusters could have been included by AGREEMENT and any singleton at $t_s$ that

joins $C_{t_S+1}$ has label 0. All such $u \in C_{t_s}$ as well. Then we have

$$\begin{aligned}
|C_{t_s} \setminus C_{t_s+1}| &\geq |C_{t_s} \setminus Z^{start}| \\
&\geq |C_{t_s}| - |Z^{start}| \\
&\geq |C_{t_s}| - 100\epsilon |C_{t_s+1}| \\
&\geq |C_{t_s}| - 300\epsilon |C_{t_s}| \\
&\geq (1 - 300\epsilon)|C_{t_s}|
\end{aligned}$$

where the first inequality is due to $Z^{start} \subseteq C_{t_s+1}$, the second due to lower bounding the cardinality of set difference by the difference of set cardinalities, the third due to the assumption that $E_{t_s+1}$ is true and the fourth due to Lemma 18. Notice that this is a contradiction to Eq. (\*) as for adequately small $\epsilon$ we have $1 - 300\epsilon \geq 18\epsilon$ and we obtain the statement.

**Observation 4** Now we attempt to study the iterations before $t^*$, in which there might have been multiple transition events. From Lemma 17 we know that between any two consecutive transition events, there is an important event. Furthermore, the lemma states that between the important event and the second transition event, there are no split events.

**Observation 5** We show the claim by arguing that there is no $u \in Z_{C_{t^*}} : s(u,t) = 0$, i.e. all such $u$ have been singletons sometime in the past. In that direction, we consider two cases. If there was another transition event before $t^*$ we denote that $t^{**}$ and from observation 5 we know that there must have been an important event $t_I \in \{t^{**}, \dots, t^*\}$ and after that no split events until $t^*$. Otherwise, there is no transition event before $t^*$. But any $u \in C_{t_C}$ was singleton at $t_C - 1$ so $Z^s_{C_{t_C}} = Z_{C_{t_C}}$ and the first split event after $t_C$ happens at $t^*$. As we have already shown in observation 2, the absence of split events in consecutive iterations implies that in the beginning of iteration $t^*$ we have $Z^s_{C_{t^*}} = Z_{C_{t^*}} > 100\epsilon |C_{t^*}|$ by definition of $t^*$ as a transition event. Then we have $|\{u \in C_{t^*} : s(u,t^*) = 0\}| < |C_{t^*} \setminus Z^s_{C_{t^*}}| \leq (1 - 100\epsilon)|C_{t^*}|$.

**Putting everything together** From the initial assumption and observation 3 we have $|\{u \in C_t : s(u,t) = 0\}| \geq |Z_{C_t} \setminus Z^s_{C_t}| \geq (1 - \frac{\epsilon}{100})|C_t|$ and from observation 5 we have that $|\{u \in C_{t^*} : s(u,t^*) = 0\}| < (1 - 100\epsilon)|C_{t^*}|$. Now we define $N = |\{u \in C_t \setminus C_{t^*} : s(u,t) = 0\}|$, i.e. the nodes that have never been singleton, belong to $C_t$ and were not present at time $t^*$. By the definition of $N$ we have:

$$\begin{aligned}
|N| &\geq |\{u \in C_t : s(u,t) = 0\}| - |\{u \in C_{t^*} : s(u,t^*) = 0\}| \\
&= (1 - \frac{\epsilon}{100})|C_t| - (1 - 100\epsilon)|C_{t^*}| \\
&\geq (1 - \frac{\epsilon}{100})|C_t| - \frac{1 - 100\epsilon}{1 - 20\epsilon}|C_t| \\
&\geq (1 - \frac{\epsilon}{100})|C_t| - (1 - \frac{80\epsilon}{1 - 20\epsilon})|C_t| \\
&\geq (80\epsilon - \frac{\epsilon}{100})|C_t| \\
&\geq 79\epsilon |C_t| \quad\quad\quad\quad\quad\quad\quad\quad\quad\quad\quad\quad\quad\quad (*)
\end{aligned}$$

where the second inequality is due to Lemma 16 and the last inequality holds since $\epsilon$ is sufficiently small. Notice that any node in $N$ must have been included in some cluster between $t^*$ and $t$ at the

iteration when it appeared in the input, otherwise, it would have been a singleton before. Consider the first such node $u \in N$ to be included in the history of $C$ and denote the iteration when it appeared in the input $t_u$. Further define $N^u = N \setminus \{u\}$. In the following, we show that its degree increases multiplicatively between $t_u$ and $t$ which implies an important event and thus concludes the proof. We start by lower bounding the nodes of $N^u$ that are neighbors of $u$ at $t$ and we have:

$$
\begin{aligned}
|N_{G_t}(u) \cap N^u| &= |N^u \setminus (N_{G_t}(u) \setminus N^u)| \\
&\geq |N^u| - |N_{G_t}(u) \setminus N^u| \\
&\geq |N^u| - |N_{G_t}(u) \setminus C_t| \\
&\geq 79\epsilon|C_t| - 1 - \frac{3\epsilon}{1-3\epsilon}|C_t| \\
&\geq 70\epsilon|C_t| - 1
\end{aligned}
\tag{1}
$$

where the third inequality is due to Eq. (*), Property 10 and the fact that $1-3\epsilon \geq 1/3$ for adequately small $\epsilon$. Then we lower bound $C_t$ by the neighbors of $u$ at time $t_u$:

$$
\begin{aligned}
|C_t| &\geq (1 - 20\epsilon)|C_{t_u}| \\
&\geq (1 - 20\epsilon)(1 - 3\epsilon)|N_{G_{t_u}}(u)| \\
&\geq (1 - 23\epsilon)|N_{G_{t_u}}(u)|
\end{aligned}
\tag{2}
$$

where the first inequality is due to Lemma 16, the second inequality is due to Property 3 and the third inequality due to omitting square terms. The increase in the degree of $u$ is:

$$
|N_{G_t}(u)| - |N_{G_{t_u}}(u)| = |N_{G_t}(u) \setminus N_{G_{t_u}}(u)| \geq |N_{G_t}(u) \cap N^u| \geq 70\epsilon|C_t| - 1
$$

where the first inequality is by definition of $N^u$ and the second is due to Eq. (1). Reordering the terms we have:

$$
\begin{aligned}
|N_{G_t}(u)| &\geq |N_{G_{t_u}}(u)| + 70\epsilon|C_t| - 1 \\
&\geq |N_{G_{t_u}}(u)| + 70\epsilon(1 - 23\epsilon)|N_{G_{t_u}}(u)| - 1 \\
&\geq (1 + 35\epsilon)|N_{G_{t_u}}(u)| - 1 \\
&\geq (1 + 35\epsilon)|N_{G_{t_u}}(u)| - 2\epsilon|N_{G_t}(u)|
\end{aligned}
$$

where the second inequality is due to Eq. (2), the third due to $1 - 23\epsilon \geq 1/2$ for adequately small $\epsilon$ and the fourth due to the assumption that $|N_{G_t}(u)| \geq \frac{1}{2\epsilon}$. Reordering again we have that:

$$
|N_{G_t}(u)| \geq \frac{1 + 35\epsilon}{1 + 2\epsilon}|N_{G_{t_u}}(u)| \geq (1 + 32\epsilon)|N_{G_{t_u}}(u)|
$$

where $\frac{1+35\epsilon}{1+2\epsilon} \geq 1 + 32\epsilon$ holds for small enough epsilon. By definition we know that $|N_{G_{t_u}}(u)| \geq |N_{G_{t_{u_{C,t}}}}(u_{C,t})|$ and since $u, u_{C,t} \in C$, Property 8 gives $|N_{G_t}(u_{C,t})| \geq (1 - 5\epsilon)|N_{G_t}(u)|$. Overall we have that:

$$
|N_{G_t}(u_{C,t})| \geq (1 - 5\epsilon)|N_{G_t}(u)| \geq (1 - 5\epsilon)(1 + 32\epsilon)|N_{G_{t_u}}(u)| \geq (1 + 26\epsilon)|N_{G_{t_{u_{C,t}}}}(u_{C,t})|
$$

where $(1 - 5\epsilon)(1 + 32\epsilon) \geq 1 + 26\epsilon$ for small enough $\epsilon$. This concludes our contradiction as it implies that there was an important on iteration $t$ in which $u$ participated.

∎

**Lemma 16** *Let $C_{t_1}, \ldots, C_{t_l}$ be the "history" of cluster $C_{t_l}$. Then for any $i, j \in \{1, \ldots, l\}$ we have that if $i > j$ then $|C_{t_i}| \geq (1 - 20\epsilon)|C_{t_j}|$.*

**Proof**    We consider two cases, depending on whether the intersection $C_{t_i} \cap C_{t_j}$ is empty.

1. Let $u \in C_{t_i} \cap C_{t_j} \neq \emptyset$. Then we have:

$$|C_{t_i}| \overset{Property3}{\geq} (1 - 3\epsilon)|N_{G_{t_i}}(u)| \overset{t_i > t_j}{\geq} (1 - 3\epsilon)|N_{G_{t_j}}(u)|$$

$$\overset{Property4}{\geq} (1 - 3\epsilon)(1 - 9\epsilon)|C_{t_j}| \geq (1 - 12\epsilon)|C_{t_j}|$$

2. If $C_{t_i} \cap C_{t_j} = \emptyset$ we assume towards a contradiction that $|C_{t_i}| < (1 - 20\epsilon)|C_{t_j}|$ and w.l.o.g. we assume that $t_j$ is the maximum $t \leq t_i$, for which this condition holds. In other words, for all $t_{j'} \in (t_j, t_i]$ we have that $|C_{t_i}| \geq (1 - 20\epsilon)|C_{t_{j'}}|$. We end up in a contradiction by proving that $C_{t_i} \cap C_{t_j} = \emptyset$ and the fact that $C_{t_i}, C_{t_j}$ both belong to the same "history" sequence implies that $\exists t_{j'} \in (t_j, t_i]$ such that $|C_{t_j}| \leq |C_{t_{j'}}|$. Since $C_{t_i} \cap C_{t_j} = \emptyset$ let $t_{j'}$ be the first time after $t_j$ such that $C_{t_{j'}} \cap C_{t_j} = \emptyset$. We have that:

$$C_{t_{j'-1}} \cap C_{t_j} \neq \emptyset \text{ and } C_{t_{j'}} \cap C_{t_j} = \emptyset \text{ by the definition of } t_{j'}$$

$$C_{t_{j'}} \cap C_{t_{j'-1}} \neq \emptyset \text{ since they are consecutive in the "history" sequence}$$

Let $u_1 \in C_{t_j} \cap C_{t_{j'-1}}$ and $u_2 \in C_{t_{j'-1}} \cap C_{t_{j'}}$, we prove that $|C_{t_j}| \leq |C_{t_{j'}}|$ by upper and lower bounding $|N_{G_{t_{j'-1}}}(u_2) \cap C_{t_j}|$. For the upper bound we have:

$$|N_{G_{t_{j'-1}}}(u_2) \cap C_{t_j}| \overset{j'>j'-1}{\leq} |N_{G_{t_{j'}}}(u_2) \cap C_{t_j}| \overset{C_{t_j} \cap C_{t_j} = \emptyset}{\leq} |N_{G_{t_{j'}}}(u_2) \setminus C_{t_{j'}}| \overset{Property10}{\leq} \frac{3\epsilon}{1 - 3\epsilon}|C_{t_{j'}}|$$

For the lower bound we have:

$$|N_{G_{t_{j'-1}}}(u_2) \cap C_{t_j}| \geq |N_{G_{t_{j'-1}}}(u_1) \cap C_{t_j}| - |N_{G_{t_{j'-1}}}(u_2) \setminus N_{G_{t_{j'-1}}}(u_1)|$$

$$\overset{\substack{Property7 \text{ since both } u_1, u_2 \in C_{t_{j'-1}}}}{\geq} |N_{G_{t_{j'-1}}}(u_1) \cap C_{t_j}| - 5\epsilon|N_{G_{t_{j'-1}}}(u_2)|$$

$$\overset{j'>j'-1 \geq j}{\geq} |N_{G_{t_j}}(u_1) \cap C_{t_j}| - 5\epsilon|N_{G_{t_{j'}}}(u_2)|$$

$$\overset{Property6}{\geq} (1 - 9\epsilon)|C_{t_j}| - 5\epsilon|N_{G_{t_{j'}}}(u_2)|$$

$$\overset{Property3}{\geq} (1 - 9\epsilon)|C_{t_j}| - \frac{5\epsilon}{1 - 3\epsilon}|C_{t_{j'}}|$$

Combining the upper and lower bounds of $|N_{G_{t_{j'-1}}}(u_2) \cap C_{t_j}|$ we get:

$$\frac{3\epsilon}{1 - 3\epsilon}|C_{t_{j'}}| \geq (1 - 9\epsilon)|C_{t_j}| - \frac{5\epsilon}{1 - 3\epsilon}|C_{t_{j'}}| \Rightarrow$$

$$\frac{8\epsilon}{1 - 3\epsilon}|C_{t_{j'}}| \geq (1 - 9\epsilon)|C_{t_j}| \Rightarrow$$

$$|C_{t_{j'}}| \geq \frac{(1 - 3\epsilon)(1 - 9\epsilon)}{8\epsilon}|C_{t_j}| \xrightarrow{\epsilon \text{ small enough}}$$

$$|C_{t_{j'}}| \geq |C_{t_j}|$$

which is a contradiction.

■

**Lemma 17** *Let $C \in \mathcal{C}_t^{agr}$ and $t_1, t_2$ where $t_C \leq t_1 < t_2 \leq t$ be two consecutive transition events in the history of $C$. Then we have the following:*

1. *$t_2 \geq t_1 + 2$.*

2. *There exists at least one $t_I \in \{t_1 + 1, \ldots, t_2 - 1\}$ so that there is an important event at $t_I$.*

3. *For every $t' \in \{t_I, t_2 - 1\}$ we have $E_{t'}$ true, i.e. no split events.*

**Proof** We proceed to prove each statement:

1. At time $t_1, t_2$ we have transition events which implies $\overline{E_{t_1}} \wedge E_{t_2-1} \wedge \overline{E_{t_2}}$ is true by definition. Therefore $t_2 - 1 > t_1$ and the statement follows.

2. Assume there is no such $t_I$ towards a contradiction. After the transition event at time $t_1$, we have $Z_{C_{t_1}} = C_{t_1}$ due to the split event. By induction we show that for any $t' \in \{t_1, \ldots, t_2 - 1\}$ we have $E_{t'}$ false, i.e. consecutive split events. The base case $t' = t_1$ is implied by $t_1$ being a transition event. For the inductive step, we know that at the end of the split event at time $t'$ we have $Z_{C_{t'}} = C_{t'}$, which we denote by $Z'$. Since $C_{t'}, C_{t'+1}$ are consecutive clusters in the history of $C$ by Lemma 18 we have:

$$|C_{t'+1} \setminus C_{t'}| \leq \frac{2}{3}|C_{t'+1}|. \tag{*}$$

Now we examine the size of $Z_{C_{t'+1}}$ before line 2 of Algorithm 4, which we denote $Z'_{next}$. Note that any node $u \in Z'$ cannot exit the cluster between consecutive iterations unless AGREEMENT sets it as singleton which implies $Z'_{next} \subseteq Z' = C_{t'}$ and cannot change its label $g(u)$ from 0 to 1 as we have assumed no important events. Then all nodes $v \in C_{t'+1} : g(v) = 1$ must be nodes that were not in $C_{t'}$ and we have:

$$|C_{t'+1} \setminus C_{t'}| \geq |C_{t'+1} \setminus Z'_{next}| = |C_{t'+1}| - |Z'_{next}|.$$

Now assume $E_{t'+1}$ is true. That would imply that $Z'_{next} \leq 100\epsilon|C_{t'+1}|$ and from the previous inequality we have:

$$|C_{t'+1} \setminus C_{t'}| \geq (1 - 100\epsilon)|C_{t'+1}|$$

which is a contradiction to Eq. (*) and concludes the inductive step. Overall we have shown $E_{t_2-1}$ is false which is a contradiction to $t_2$ being a transition event.

3. At time $t_I$ we showed that there is an important event, so we know that $E_{t_I}$ is true as there could not have been a split event. Suppose that there is some $t' \in \{t_I + 1, t_2 - 1\}$ such that $E_{t'}$ is false. That implies that there must have been a transition event at some time $t'' \in \{t_I + 1, t'\}$ which contradicts the fact that $t_1, t_2$ are consecutive.

■

**Lemma 18** *Suppose $C \in \mathcal{C}_t^{agr}$ and $C_{t'}, C_{t'+1}$ are two consecutive clusters in the history of $C$ at times $t', t'+1$ respectively. Then for adequately small $\epsilon$ we have the following:*

1. *$|C_{t'}| \geq \frac{1}{3}|C_{t'+1}|$,*

2. *$|C_{t'} \setminus C_{t'+1}| \leq 18\epsilon|C_{t'}|$, and*

3. *$|C_{t'+1} \setminus C_{t'}| \leq \frac{2}{3}|C_{t'+1}|$.*

**Proof** By definition of the history of a cluster, there exists $u \in C_{t'} \cap C_{t'+1} \neq \emptyset$. Using the properties of AGREEMENT we proceed to prove the statements in the same order:

1. Note that

$$
\begin{aligned}
|C_{t'}| &\geq |C_{t'} \cap N_{G_{t'}}(u)| \\
&\geq (1 - 3\epsilon)|N_{G_{t'}}(u)| \\
&\geq (1 - 3\epsilon)(|N_{G_{t'+1}}(u)| - 1) \\
&\geq (1 - 3\epsilon)(1 - 9\epsilon)|C_{t'+1}| - (1 - 3\epsilon) \\
&\geq (1 - 12\epsilon)|C_{t'+1}| - (1 - 3\epsilon)
\end{aligned}
$$

where the first inequality is due to the intersection, the second due to [Property 1], the third since the neighbors of $u$ cannot increase by more than 1 from $t'$ to $t'+1$, the fourth due to [Property 6] and the fifth by omitting the positive square terms. Note that $|C_{t'+1}| \geq 2$ by definition of a cluster which implies $1 - 3\epsilon \leq \frac{1-3\epsilon}{2}|C_{t'+1}|$ and we have:

$$
|C_{t'}| \geq (1 - 12\epsilon)|C_{t'+1}| - \frac{1 - 3\epsilon}{2}|C_{t'+1}| = \frac{1 - 21\epsilon}{2}|C_{t'+1}| \geq \frac{1}{3}|C_{t'+1}|
$$

where the last inequality holds for $\epsilon \leq \frac{1}{63}$.

2. Note that

$$
\begin{aligned}
|C_{t'} \setminus C_{t'+1}| &= |(C_{t'} \setminus C_{t'+1}) \setminus N_{G_{t'+1}}(u)| + |N_{G_{t'+1}}(u) \cap (C_{t'} \setminus C_{t'+1}| \\
&\leq |(C_{t'} \setminus C_{t'+1}) \setminus N_{G_{t'}}(u)| + |(N_{G_{t'+1}}(u) \cap C_{t'}) \setminus C_{t'+1}| \\
&\leq |C_{t'} \setminus N_{G_{t'}}(u)| + |N_{G_{t'+1}}(u) \cap C_{t'} \setminus C_{t'+1}|
\end{aligned}
$$

where the in the first line we partition the set using $N_{G_{t'+1}}(u)$, in the first inequality we use that $N_{G_{t'}}(u) \subseteq N_{G_{t'+1}}(u)$ and reorder the second term, in the second inequality we use that $N_{G_{t'+1}}(u) \cap C_{t'} \subseteq N_{G_{t'+1}}(u)$. Now using [Property 5] we can upper bound the first term:

$$
|C_{t'} \setminus N_{G_{t'}}(u)| \leq 9\epsilon|C_{t'}|.
$$

For the second term, we proceed as follows:

$$
\begin{aligned}
|N_{G_{t'+1}}(u) \cap C_{t'} \setminus C_{t'+1}| &\leq |N_{G_{t'+1}}(u) \setminus C_{t'+1}| \\
&\leq 3\epsilon|N_{G_{t'+1}}(u)| \\
&\leq 3\epsilon(|N_{G_{t'}}(u)| + 1) \\
&\leq \frac{3\epsilon}{1 - 3\epsilon}|C_{t'}| + 3\epsilon \\
&\leq 9\epsilon|C_{t'}|
\end{aligned}
$$

where the second inequality is due to Property 2, the third inequality is due to $N_{G_{t'}}(u) \le N_{G_{t'+1}}(u)+1$ since $u$ can acquire at most 1 new neighbor between consecutive times (exactly the new node arriving in the input), the fourth due to Property 10 and the fifth due to $1 - 3\epsilon \ge 0.5$ as $\epsilon$ is adequately small. Then combining the two bounds we have the statement.

3. The proof is similar to the previous case so we state it briefly:

$$
\begin{aligned}
|C_{t'+1} \setminus C_{t'}| &\le |(C_{t'+1} \setminus C_{t'}) \setminus N_{G_{t'+1}}(u)| + |N_{G_{t'+1}}(u) \cap (C_{t'+1} \setminus C_{t'}| \\
&\le |C_{t'+1} \setminus N_{G_{t'+1}}(u)| + |N_{G_{t'+1}}(u) \setminus C_{t'}| \\
&\le 9\epsilon|C_{t'+1}| + 1 + |N_{G_{t'}}(u) \setminus C_{t'}| \\
&\le 9\epsilon|C_{t'+1}| + 1 + 3\epsilon|N_{G_{t'}}(u)| \\
&\le 9\epsilon|C_{t'+1}| + 1 + 3\epsilon|N_{G_{t'+1}}(u)| \\
&\le 9\epsilon|C_{t'+1}| + 1 + \frac{3\epsilon}{1 - 3\epsilon}|N_{G_{t'+1}}(u)| \\
&\le 9\epsilon|C_{t'+1}| + 1 + 6\epsilon|N_{G_{t'+1}}(u)| \\
&\le (15\epsilon + 1/2)|C_{t'+1}|
\end{aligned}
$$

which gives the statement as for small enough $\epsilon$ $15\epsilon + 1/2 \le 2/3$ and the previous inequality is due to $|C_{t'+1}| \ge 2$ by definition of the cluster.

∎

## Appendix E. Clusters with Small Degrees

In this section we lift the assumption that all nodes have degree at least $\frac{1}{2\epsilon}$. We start by defining the subset of clusters of $\mathcal{C}_t^{agr}$ whose nodes have low degrees:

$$
\mathcal{C}_t^s = \{C \in \mathcal{C}_t^{agr} : |N_{G_t}(u)| < \frac{1}{\epsilon}, \ \forall u \in C\}.
$$

Notice that for any cluster $C \in \mathcal{C}_t^{agr} \setminus \mathcal{C}_t^s$, there exists a node $u \in C$ such that $|N_{G_t}(u)| \ge \frac{1}{\epsilon}$, therefore by Property 8, every other node $v \in C$ has $|N_{G_t}(v)| \ge \frac{1-5\epsilon}{\epsilon} \ge \frac{1}{2\epsilon}$ and our proofs so far hold. We complement our analysis so far by showing that the clusters of $\mathcal{C}_t^s$ do not violate any of our results.

**Lemma 19** *The recourse of any node $u$ is $O\left(\frac{\log n}{\epsilon}\right)$.*

**Proof** Fix iteration $T$ and define the first iteration when $u$ was in a cluster of $\mathcal{C}_T^{agr} \setminus \mathcal{C}_T^s$:

$$
t_f = \min\{t' \le T : u \in \bigcup_{C \in \mathcal{C}_{t'}^{agr} \setminus \mathcal{C}_{t'}^s} C\}.
$$

We split the analysis into the following cases:

1. For $t \in [t_f, T]$ we have large degrees so our previous analysis suffices.

2. For $t \in [1, t_f - 1]$, we show that $u$ can only incur $O\left(\frac{1}{\epsilon}\right)$ recourse.

First, for $t \in [t_f, T]$, node $u$ has $|N_{G_t}(u)| \geq \frac{1-5\epsilon}{\epsilon}$ because it belongs in cluster $C \in \mathcal{C}_t^{agr} \setminus \mathcal{C}_t^s$. Therefore, for any such iteration $t$, node $u$ belongs in clusters $C \in \mathcal{C}_t^{agr}$ such that any $v \in C$ must have $|N_{G_t}(v)| \geq \frac{(1-5\epsilon)^2}{\epsilon} \geq \frac{1-10\epsilon}{\epsilon} \geq \frac{1}{2\epsilon}$ by Property 8, so its recourse is $O\left(\frac{\log n}{\epsilon}\right)$ by Theorem 1. Second, for iterations $t \in [1, t_f - 1]$, $u$ can only incur recourse due to one of the three subroutines of our algorithm. We claim the following:

1. AGREEMENT can only contribute at most 2 in the recourse of $u$.

2. MAKECONSISTENT cannot contribute to the recourse of $u$.

3. MAKEFAIR can only contribute at most $O\left(\frac{1}{\epsilon}\right)$ to the recourse of $u$.

For the first point, note that due to Lemma 21 $u$ is either singleton or belongs to a clique in $\mathcal{C}_t^s$. Then, any recourse incurred due to AGREEMENT is because $u$ transitioned from one to the other. We argue that it can only transition at most twice, once from being singleton to being added to a clique $C$ and a second one from a clique $C$ to singleton. We now prove that if a node is in a clique $C \in \mathcal{C}_{t-1}^s$ at time $t-1$ and is clustered as a singleton at time $t$ then it will be clustered as a singleton by AGREEMENT for all times in $[t, t_f - 1]$. Let $w$ be the node that arrives at time $t$. If $N_{G_t}(w) = C$ or $N_{G_t}(w) \cap C = \emptyset$ then $u$ will be clustered in a non-singleton cluster at time $t$. Thus, if $u$ is a singleton at time $t$, the only remaining case is that $N_{G_t}(w) \neq C$ and $N_{G_t}(w) \cap C \neq \emptyset$. This implies that $N_{G_t}(w) \neq N_{G_t}(u)$, therefore, $u$ cannot be part of any isolated clique $C' \in \mathcal{C}_{t'}^s$ for all $t' \in [t, t_f - 1]$ and it is a singleton $\{u\} \in \mathcal{C}_{t'}^s$. Overall these cases imply that the recourse of node $u$ due to AGREEMENT is at most 2.

For the second point, note that due to Lemma 22, if $C \in \mathcal{C}_t^s$ then $C \in \widetilde{C}_t$, so MAKECONSISTENT contributes 0 to the recourse of $u$.

For the third point, assume that on iteration $t$, $u$ belongs in a non-singleton clique $C \in \mathcal{C}_t^s$. Obviously, if $u$ is a singleton, MAKEFAIR will not affect it. Otherwise, if $C \in \mathcal{C}_t^s \cap \mathcal{C}_{t+1}^s \cap \mathcal{C}_t$, then $C \in \mathcal{C}_{t+1}$ as well, because on iteration $t+1$, $C_{prev} = C$ and the second if-statement of Algorithm 3 has its condition true. Therefore, if $C \in \mathcal{C}_t^s \cap \mathcal{C}_t \setminus \mathcal{C}_{t+1}$ then we can conclude that $C \notin \mathcal{C}_{t+1}^s$. This can either happen because a new node $w$ appeared in the input on iteration $t+1$, due to which $C$ is no longer a clique, or because $w$ was added to $C$ and MAKEFAIR split the cluster. Notice that in this second case, the degree of $u$ increased by 1 and so did its recourse. This can only happen at most $\frac{1}{2\epsilon}$ times, until $t \geq t_f$. ∎

**Lemma 20** *For a constant $\epsilon$ small enough we have* $\mathrm{cost}(\widetilde{\mathcal{C}}_t) \leq \Theta\left(\frac{1}{\epsilon^3}\right) \cdot \mathrm{cost}(\mathcal{O}_t)$.

**Proof** We consider all edges that might have contributed to $\mathrm{cost}(\widetilde{\mathcal{C}}_t)$ and proceed to complete the analysis even for the case of nodes with low degree:

1. All edges $(u, v)$ such that $u, v \in \bigcup_{C \in \mathcal{C}_t^s} C$ can be charged to $\mathrm{cost}(\mathcal{C}_t^{agr})$. First, notice that either $\{u\}, \{v\} \in \mathcal{S}(\mathcal{C}_t^s)$, or $u, v \in C$ where $C \in \mathcal{C}_t^s$ is a clique by Lemma 21. In the first case, $(u, v)$ is paid by $\mathcal{C}_t^{agr}$. In the second case, we have that $u, v \in C$ and by Lemma 22, MAKECONSISTENT does not split $C$, so $(u, v)$ contributes cost 0 to $\mathrm{cost}(\widetilde{\mathcal{C}}_t)$.

2. All edges $(u, v)$ such that $u \in \bigcup_{C \in \mathcal{C}_t^{agr} \setminus \mathcal{C}_t^s} C$, $v \in \bigcup_{C \in \mathcal{C}_t^s} C$, are paid by AGREEMENT because $v$ must be a singleton in $\mathcal{C}_t^{agr}$ by Lemma 21 and therefore also a singleton in $\widetilde{\mathcal{C}}_t$.

3. All other edges $(u, v)$ can be charged to $\text{cost}(\mathcal{O}_t)$ due to Lemma 5, since $|N_{G_t}(u)|, |N_{G_t}(v)| \geq \frac{1}{2\epsilon}$.

∎

**Lemma 21** *Let $C \in \mathcal{C}_t^s$ with $|C| \geq 2$, then $C = N_{G_t}(u)$ for all $u \in C$, i.e. $C$ is an isolated clique in $G_t$.*

**Proof** Let $u, v$ be two distinct nodes in $C$ that are in agreement. By the agreement definition we have:

$$|N_{G_t}(u) \triangle N_{G_t}(v)| < \epsilon \max\{|N_{G_t}(u)|, |N_{G_t}(v)|\} < \epsilon \cdot \frac{1}{\epsilon} = 1$$

where the second inequality holds due to the definition of $\mathcal{C}_t^s$. The latter implies that all nodes in $C$ which are in agreement must have the same neighborhood in $G_t$. We continue arguing that all nodes in $C$ have the same neighborhood. Note that in any non-singleton cluster $C$ there must be a node $u \in C$ which is heavy. Since, $|N_{G_t}(u)| < \frac{1}{\epsilon}$ for $u$ to be heavy it must be that it is in $\epsilon$-agreement with at least a $(1 - \epsilon)$ fraction of its neighbors, i.e.,

$$(1 - \epsilon)|N_{G_t}(u)| = |N_{G_t}(u)| - \epsilon|N_{G_t}(u)| > |N_{G_t}(u)| - 1$$

due to $|N_{G_t}(u)| < \frac{1}{\epsilon}$, which implies that $u$ is in $\epsilon$-agreement with all of its neighbors. Thus, $\forall v \in N_{G_t}(u)$ we have $N_{G_t}(u) = N_{G_t}(v)$. i.e., $N_{G_t}(u)$ forms an isolated clique which is equal to $C$. ∎

**Lemma 22** *Let $C \in \mathcal{C}_t^s$ be a cluster such that $|C| \geq 2$, then $C \in \widetilde{\mathcal{C}}_t$.*

**Proof** We define the first and last time that $C \in \mathcal{C}_t^s$ as follows:

$$t_f = \min\{t' \leq t : C \in \mathcal{C}_{t'}^s\}, \quad t_l = \max\{t' \geq t : C \in \mathcal{C}_{t'}^s\}.$$

We proceed to show that for any $|C| \geq 2$ and for all $t' \in [t_f, t_l]$ we have that (i) $C \in \mathcal{C}_{t'}^s$ and (ii) $C \in \widetilde{\mathcal{C}}_{t'}$.

For part (i), we show that for all $t' \in [t_f, t_l]$ we have that $C \in \mathcal{C}_{t'}^s$, i.e. AGREEMENT does not split $C$ between the first and the last iteration when is created. Note that $C$ must be a clique in any iteration $t'$ such that $C \in \mathcal{C}_{t'}^s$ by Lemma 21 and we proceed by contradiction. Assume that there exists $t' \in [t_f, t_l]$ such that $C \notin \mathcal{C}_{t'}^s$ and define the first such iteration $t_s = \min\{t' \in [t_f, t_l] : C \notin \mathcal{C}_{t'}^s\}$. Then the node $u$ that appeared in the input at $t_s$, must have had $N_{G_{t_s}}(u) \neq C, N_{G_{t_s}}(u) \cap C \neq \emptyset$, otherwise either $N_{G_{t_s}}(u) \cap C = \emptyset$ in which case $C \in \mathcal{C}_{t_s}^s$ and the contradiction is by definition of $t_s$, or $N_{G_{t_s}}(u) = C$ in which case $C \cup \{u\} \in \mathcal{C}_t^{agr}$ so $t_s > t_l$ as $C \notin \mathcal{C}_{t''}^s$ for any $t'' \geq t_s$ since $C$ is no longer a clique. But since $N_{G_{t_s}}(u) \subset C$, there exist $v, w \in C$ such that $v \in N_{G_{t_s}}(u)$ and $w \notin N_{G_{t_s}}(v)$. Then, once again we must have that $t_s > t_l$ since $C$ is not a clique anymore, which concludes the contradiction.

For part (ii), we show that $g_w = 1$ for all $w \in C$, at the end of any iteration $t' \in [t_f, t_l]$, which suffices, by definition of the labels $g$. We proceed by induction on $|C| = k \geq 2$. For the base case, $k = 2$, $C = \{u, v\} \in \mathcal{C}_{t'}^s$ which implies $N_{G_{t'}}(u) = N_{G_{t'}}(v) = C$ and wlog $v$ arrived after $u$. Let $t_v$ be the iteration when $v$ arrived. At the start of $t_v$, we have that $g(u) = 0$ since it had no edges in iteration $t_v - 1$, therefore it was singleton. On iteration $t_v$, $|N_{G_{t_v}}(u)| = 2 = 2 \cdot 1 = 2 \cdot |N_{G_{t_v-1}}(u)| > (1 + 20\epsilon)|N_{G_{t_v-1}}(u)|$, therefore we have an important event for $C$ and $g(u)$ is set to 1. Overall $t_f = t_v$ and for all $t' \in [t_f, t_v]$, at the start of iteration $t'$, $g(u) = g(v) = 1$, so none of the two if statements of MAKECONSISTENT are true and $C$ is not split. For the inductive step, assume we have the statement for $k \leq k_0$ and consider $k = k_0 + 1$. As before, let $v \in C$ be the node that arrives last in the input, among the nodes of $C$, on iteration $t_v$. Since $C$ is a clique, then $C \setminus \{v\}$ must also be a clique of size $k_0$. Then by inductive hypothesis $g(u) = 1$ for all $u \in C \setminus \{v\}$ and $v \in C \in \mathcal{C}_t^s$ therefore at the start of MAKECONSISTENT on $t_v$, $g(v) = 1$. Once again, since all nodes $u \in C$ have $g(u) = 1$, the cluster is not split.

∎

