# OpenReview forum: "Cost-Free Fairness in Online Correlation Clustering"
_algorithmiclearningtheory.org/ALT/2025/Conference — ALT 2025_

### Official Review · Reviewer_eEcB · 2024-10-29
**Solid work**

**Rating:** 7
**Confidence:** 2

**Review:**

This paper considers the problem of fair clustering, in which each vertex is assigned at least one of $\ell$ colours, and each cluster $C$ containing more than one vertex is required to satisfy  $|C\cap V_i| \leq a_i|C|$ for all $i$, where $V_i$​ denotes the set of vertices of the $i$-th colour. This paper presents an algorithm that achieves a constant-factor cost approximation with fairness constraints throughout the reading of the input stream of the graph, where each vertex is given one by one.

The paper achieves both fairness and online guarantees with tight recourse. Some core algorithmic procedures are based on the previous work of Cohen-Addad et al. on online correlation clustering, but this submission simplifies the analysis in that paper.

Overall, this paper is a solid work in online clustering, and I recommend its acceptance.

Questions:
- The introduction begins by discussing the application of fairness, but fairness in clustering specifically is not mentioned. Please provide some background on fairness in clustering to better motivate the problem.
- Is there a guarantee on the number of clusters created by the algorithm?
- It seems to me that the recourse of Algorithm 1 is $O(\log n/\epsilon)$, where the $1/\epsilon$ factor comes from the logarithm base of $1+\epsilon$. Is it assumed here that $\epsilon$ is a constant? Otherwise the dependence on $\epsilon$ cannot be omitted.
- What would the dependence on epsilon be in Lemma 5 if epsilon is not constant?
- In the main theorem stated on Page 2, what is the dependence of recourse and approximation ratio on a_i and rho? Many things seem to be ignored by assuming that the parameters are constants.

Minor comments:
- Page 2, Theorem statement, last line: $\rho$-fair has not been defined at this point. In fact it is not defined throughout the paper.
- Page 8, Definition 1, second line: Should $N_{G_{t_{u_{C,t}}}}$ be $N_{G_{\tilde{t}_{u_{C,t}}}}$?
- Page 8, line after Lemma 3: providing -> provide and “Lemma 2in Appendix B” -> “Lemma 2 to Appendix B”.
- Page 8, proof of Theorem 1, line 4: delete “equal to”
- Page 8, Line -5: “there \exists” -> “there exist”
- Page 9, $C_t^{agreement}$ should be $C_t^{agr}$
- Page 9, Line 1: “\forall t” -> “for all t”
- Page 9, Line 13: what does “\simeq” mean?
- Page 10, Theorem 4 statement: Is the number of colours denoted by $\ell$ or $L$? Please be consistent.
- Page 12, Line 6: should “$k+1$ clusters” be “the $(k+1)$-st cluster”?
- Page 12, Line -11: belong -> belong to
- Page 12, Line -5: in any case -> in both cases

**Paper Award:**

No

---

> ### Author Response · Authors · 2024-11-24
>
> We thank the reviewer for their positive review and their time and effort in reviewing this paper.
>
> **Reviewer Questions:**
>
>
> - *The introduction begins by discussing the application of fairness, but fairness in clustering specifically is not mentioned. Please provide some background on fairness in clustering to better motivate the problem.*
>
>   **Response:** We included only citations related to fair correlation clustering. We will add a paragraph discussing general applications of fair clustering as well as relevant work.
>
> - *Is there a guarantee on the number of clusters created by the algorithm?*
>
>   **Response:** No, the correlation clustering objective is different from other common clustering objectives (e.g., $k$-means, $k$-median, $k$-center) in that it does not require a predefined number of clusters. The optimal solution might consist of a single cluster, an all-singleton solution, or any number of clusters between $1$ and $n$, depending on what minimizes the objective. This flexibility is advantageous when there is no a priori information about the number of clusters that best represents the dataset.
>
> - *It seems to me that the recourse of Algorithm 1 is $O(\log \eta / \epsilon)$, where the $1 / \epsilon$ factor comes from the logarithm base of $1 + \epsilon$. Is it assumed here that $\epsilon$ is a constant? Otherwise, the dependence on $\epsilon$ cannot be omitted.*
>
>   **Response:** Yes, $\epsilon$ is assumed to be a constant for our $O(\log n)$ bound on the recourse. For clarity, we will explicitly add the dependence on $\epsilon$ in this bound, which is indeed $O(\frac{\log n}{\epsilon})$ ($=O(\log_{1 + \epsilon} n)$) as stated by the reviewer. In our theorems, it is assumed that the fairness parameters $\alpha_i$ and $\rho$ are constants independent of the graph size, and we set $\epsilon$ to a small constant. We underline that this dependence can be avoided if we focus only on constant approximation and recourse without fairness constraints. In that case, we can modify the $(1 + 100\epsilon)$ term in the second `if`-statement of the `MakeConsistent` procedure to a constant (e.g., 2), eliminating the $\epsilon$ dependence in the recourse. However, achieving fairness under this modification is unclear. We will add a paragraph discussing these considerations and include a section on open directions.
>
> - *What would the dependence on $\epsilon$ be in Lemma 5 if $\epsilon$ is not constant?*
>
>   **Response:** If $\epsilon$ is not constant, the dependence would be $1/\epsilon^4$. The first $1/\epsilon^2$ factor arises from the agreement decomposition approximation of [1], which guided our online procedure, and the second $1/\epsilon^2$ factor comes from the `MakeConsistent` procedure. If we focus only on low recourse and constant approximation, we could modify the `MakeConsistent` procedure to eliminate the first `if`-statement, reducing the dependence on $\epsilon$ to $1/\epsilon^3$ instead of $1/\epsilon^4$. We will add a section discussing these considerations.
>
> - *In the main theorem stated on Page 2, what is the dependence of recourse and approximation ratio on $a_i$ and $\rho$? Many things seem to be ignored by assuming that the parameters are constants.*
>
>   **Response:** It is $\Theta\left(\frac{1}{\rho^5 \min_i \alpha_i^6}\right)$ for the competitive ratio and $\Theta\left( \frac{\log n}{\rho \min_i \alpha_i } \right)$ for the recourse. Let $\tilde{\epsilon}$ be a sufficiently small constant such that the agreement decomposition approximation ratio in [1] and properties in [2] (described in Appendix A) hold for all $\epsilon \leq \tilde{\epsilon}$. Previous work [1, 2] shows that such a constant exists, though it is not explicitly calculated in the latter. Setting $\epsilon = \min \left(\tilde{\epsilon}, \frac{\min_i{\alpha_i} \cdot \rho}{11200} \right)$ gives an upper bound of  $\Theta\left(\frac{1}{\epsilon^4} \frac{1}{\rho \min_i \alpha_i^2}\right)$  for the competitive ratio and $\Theta\left( \log_{1 + \epsilon} n \right)$ for the recourse. Therefore, in terms of $\alpha_i$ and $\rho$, the upper bound for the competitive ratio is  $\Theta\left(\frac{1}{\rho^5 \min_i \alpha_i^6}\right)$ and for the recourse is $\Theta\left( \frac{\log n}{\rho \min_i \alpha_i} \right)$. We plan to revise the statements of Theorems 1 and 4 by removing the assumption that $\alpha_i$ and $\rho$ are constants. Instead, we will explicitly present the exact dependence of the recourse and competitive ratio bounds on these parameters and explain how they imply the Theorem on page 2.
>
> **Response to minor comments:**
> Thank you for catching the typos and inconsistencies listed under minor comments. The reviewer is correct for all of them, and we will fix them as suggested.
>
> **References:**
>
> [1] Cohen-Addad et al. *Correlation clustering in constant many parallel rounds*. ICML 2021.
>
> [2] Cohen-Addad et al. *Online and consistent correlation clustering*. ICML 2022.

---

> > ### Comment · Reviewer_eEcB · 2024-11-26
> >
> > Thanks for the clarifications and the explanations. My evaluation remains unchanged.

---

### Official Review · Reviewer_Ntqa · 2024-11-08
**Nice extension of pre existing algorithm to deal with fairness constraints**

**Rating:** 7
**Confidence:** 3

**Review:**

The paper proposes an algorithm for online fair clustering with recourse.

Consider a graph $G=(V, E)$ the cost of a clustering $\operatorname{cost}(\mathcal{C})$ is:
$$
\operatorname{cost}(\mathcal{C})=\left|(u, v) \in E: u \not_{\mathcal{C}} v\right|+\left|(u, v) \notin E: u \sim_{\mathcal{C}} v\right|
$$
where $u \sim_{\mathcal{C}} v$ if and only if u and v ar in the same cluster of $\mathcal{C}$. The fairness constraint is that for each group $i$, the faction of group $i$ in each cluster should not exceed $a_i$, with the exception of singleton clusters that are considered fair.

The paper builds on the work of Online and consistent correlation clustering, modifying their algorithm to incorporate the fairness constraint. The crux of the modification is to slpit the clusters into singletons whevener they do not respect the fairness constraint. This is proven to not increase to much the cost of the clustering nor the number of recourses.

Overall, the paper is nice and the algorithms and proofs intuitive.

I have a few questions:
- how do the CR  and number of recourses scale with epsilon?
- why cost-free in the title? It seems that the CR  and the number of recourses scale up as epsilon decreases and the maximum value of epsilon depends on the fairness constraint. What did you mean by this?

The writing of the paper seems to have been rushed. The first half is well polished, the second more messy. Here is a non exhaustive list of possible improvements:
- page 6: give the intuition of MAKECONSISTENT before the routine.
- reintroduce AGREEMENT in the main text. It is short and will clarify things.
- Page 8, after lemma 3: the sentence is messy and is hard to parse.
- Page 7: below thm, the recouse is upper bounded by twice the latter quantity

**Paper Award:**

No

---

> ### Author Response · Authors · 2024-11-24
>
> We thank the reviewer for their positive review and their time and effort in reviewing this paper.
>
> **Reviewer Questions:**
>
> - *How do the competitive ratio (CR) and number of recourses scale with $\epsilon$?*
>   **Response:**
>   $\frac{1}{\log_2(1+\epsilon)} = O\left(\frac{1}{\epsilon}\right)$
>   for the recourse and
>   $\frac{1}{\epsilon^4} \frac{1}{\rho \min_i \alpha^2_i} \leq \frac{1}{\epsilon^6}$
>   for the competitive ratio.
>
>   To see this, let $\tilde{\epsilon}$ be a small enough constant so that the agreement decomposition approximation ratio [1] and properties [2] (as described in Appendix A) hold. We note that such a constant exists due to previous work [1, 2], but it is not calculated explicitly. Setting $ \epsilon \leq \min(\tilde{\epsilon}, \frac{\min_i \alpha_i \cdot \rho}{11200} )$ (see Theorem 1) gives us a recourse of $O \left( \log_{1+\epsilon} n \right)$. Hence the recourse scales as  $\frac{1}{\log_2(1+\epsilon)}$ with respect to $\epsilon$. The competitive ratio can be upper bounded by $\Theta \left( \frac{1}{\epsilon^4} \frac{1}{\rho \min_i \alpha^2_i} \right) \leq \Theta \left( \frac{1}{\epsilon^6} \right)$, where the first $\frac{1}{\epsilon^2}$ factor comes from the agreement decomposition approximation in [1], the second $\frac{1}{\epsilon^2}$ factor comes from the `MakeConsistent` procedure, and the last $\frac{1}{\rho \min_i \alpha_i^2}$ comes from the `MakeFair` procedure.
>
> - *Why "cost-free" in the title?*
>   **Response:**
>   The term "cost-free" in the title refers to the main implication of our result: there is no asymptotic loss in the best achievable recourse and competitive ratio guarantees when adding fairness constraints for online correlation clustering (when $\alpha_i$s and $\rho$ are constants independent of the graph size).  This observation is obtained by combining our main result together with the lower bound of [2], which shows that $\Omega(\log n)$ recourse is required for maintaining a solution that is a constant approximation. We mention the latter in the last paragraph of page 2 but will make it clearer in the next version of the paper.
>
> Thank you for your suggestions! We will incorporate them into the next version of our paper. Additionally, we plan to revise the statements of Theorems 1 and 4 by removing the assumption that $\alpha_i$s and $\rho$ are constants. Instead, we will explicitly present the exact dependence of the recourse and competitive ratio bounds on these parameters and explain how they imply the theorem on page 2.
>
> **References:**
>
> [1] Vincent Cohen-Addad, Silvio Lattanzi, Slobodan Mitrovic, Ashkan Norouzi-Fard, Nikos Parotsidis,
> and Jakub Tarnawski. *Correlation clustering in constant many parallel rounds*. ICML 2021.
>
> [2] Vincent Cohen-Addad, Silvio Lattanzi, Andreas Maggiori, and Nikos Parotsidis. *Online and consistent correlation clustering*. ICML 2022.

---

> > ### Comment · Reviewer_Ntqa · 2024-11-25
> >
> > Thank you for those clarifications. My evaluation of the paper remains unchanged.

---

### Official Review · Reviewer_N26N · 2024-11-10
**Review of Cost-Free Fairness in Online Correlation Clustering**

**Rating:** 7
**Confidence:** 4

**Review:**

Correlation clustering is a popular clustering objective, where given a graph the aim is to find a partition/clustering of the vertices such that the number of inter-cluster edges plus the number of intra-cluster non-edges is minimized. In the online version of the problem, vertices arrive sequentially one-at-a-time, and the objective is to maintain a good correlation clustering in the currently available induced subgraph for every time instance. This online version of correlation clustering is also well-studied, and algorithms are known that maintains a constant factor (eg, 0.5) competitive ratio in each time instance - while maintaining a logarithmic recourse (the number of time the cluster membership of any vertex changes).

On the other hand a ``fair'' version of correlation clustering has also been studied, where the objective is to ensure vertices of a predefined type (one of l colors) in any cluster to not be more than a certain fraction. Given a set of l fractions, a $\rho$-fair algorithm maintains the fairness constraint up to a multiplicative factor of $1+\rho$.

This paper combines these notions and asks for a *fair online* correlation clustering algorithm. It proposes an online algorithm, that is constant-factor competitive, has logarithmic recourse, and at the same time maintains the fairness constraint up to a multiplicative factor of $1+\rho$ in every step. Due to this being the optimal recourse for even a non-fair algorithm, the only scope of improvement here is the competitive ratio and $\rho$.

The algorithm seems non-trivial, because the previous online algorithms can produce far from fair algorithms. The main technical contribution is to splitting the resultant highly non-fair clusters into singletons.

Overall this seems to be a decent and timely contribution.

**Paper Award:**

No

---

> ### Author Response · Authors · 2024-11-24
>
> We thank the reviewer for their positive review and their time and effort in reviewing this paper.

---

### Meta-Review · Area_Chair_jfFJ · 2024-12-13

**Recommendation:** Accept
**Confidence:** 4

**Metareview:**

The reviewers agreed this paper was a meaningful contribution to the area of correlation clustering with recourse.

**Paper Award:**

Yes